# The elemental mechanism of transcriptional pausing

**Jason Saba[1], Xien Yu Chua[1†], Tatiana V Mishanina[1‡], Dhananjaya Nayak[1§], Tricia A Windgassen[1#], Rachel Anne Mooney[1], Robert Landick[1,2]***

[1]Department of Biochemistry, University of Wisconsin-Madison, Madison, United States; [2]Department of Bacteriology, University of Wisconsin-Madison, Madison, United States

**\*For correspondence:**
landick@bact.wisc.edu

**Present address:** [†]Department of Molecular Pharmacology, Physiology and Biotechnology, Brown University, Providence, United States; [‡]Department of Chemistry and Biochemistry, University of California, San Diego, San Diego, United States; [§]Charles River Laboratories, Raleigh-Durham, United States; [#]Department of Chemical and Systems Biology, Stanford University, Stanford, United States

**Competing interests:** The authors declare that no competing interests exist.

**Abstract** Transcriptional pausing underlies regulation of cellular RNA biogenesis. A consensus pause sequence that acts on RNA polymerases (RNAPs) from bacteria to mammals halts RNAP in an elemental paused state from which longer-lived pauses can arise. Although the structural foundations of pauses prolonged by backtracking or nascent RNA hairpins are recognized, the fundamental mechanism of the elemental pause is less well-defined. Here we report a mechanistic dissection that establishes the elemental pause signal (*i*) is multipartite; (*ii*) causes a modest conformational shift that puts γ-proteobacterial RNAP in an off-pathway state in which template base loading but not RNA translocation is inhibited; and (*iii*) allows RNAP to enter pretranslocated and one-base-pair backtracked states easily even though the half-translocated state observed in paused cryo-EM structures rate-limits pause escape. Our findings provide a mechanistic basis for the elemental pause and a framework to understand how pausing is modulated by sequence, cellular conditions, and regulators.

DOI: https://doi.org/10.7554/eLife.40981.001

## Introduction

During the first step in gene expression, transcription by RNA polymerase (RNAP) at ~30 nt/s or faster is interrupted by ≥1 s pause events every 100–200 bp (*Landick, 2006*; *Larson et al., 2014*; *Chen et al., 2015*). These pauses underlie diverse mechanisms that regulate gene expression in both prokaryotes and eukaryotes (*Figure 1A*), including attenuation, antitermination, and promoter-proximal pausing (*Jonkers and Lis, 2015*; *Zhang and Landick, 2016*; *Mayer et al., 2017*). Pausing also couples transcription to translation in bacteria or to mRNA splicing in eukaryotes (*Landick et al., 1985*; *Proshkin et al., 2010*; *Mayer et al., 2017*); defines temporal and positional windows for binding of small molecules, regulatory proteins, or regulatory RNAs to the nascent RNA transcript (*Wickiser et al., 2005*; *Artsimovitch and Landick, 2002*); mediates nascent RNA folding (*Pan et al., 1999*; *Pan and Sosnick, 2006*; *Steinert et al., 2017*); and enables termination (*Gusarov and Nudler, 1999*; *Proudfoot, 2016*). Conversely, RNA folding and the interactions of cellular molecules and complexes (e.g., ribosomes) with the elongating transcription complex (EC) modulate pausing (*Toulokhonov et al., 2001*; *Artsimovitch and Landick, 2002*; *Yakhnin et al., 2016*; *Zhang and Landick, 2016*). Despite its crucial role in cellular information processing, the biophysical mechanism of pausing remains incompletely defined.

Multiple pause mechanisms exist, but most pauses that mediate gene regulation are triggered initially by sequence-specific interactions of DNA and RNA with RNAP. An increasingly accepted view is that these initial interactions interrupt the nucleotide addition cycle by promoting entry of RNAP into a state termed the elemental pause (*Landick, 2006*), creating an elemental paused elongation complex (ePEC; *Figure 1B*). The ePEC can then rearrange into long-lived pause states by backtracking (reverse translocation of RNA and DNA), by pause hairpin (PH) formation in the RNA

**eLife digest** The information a cell needs to create a specific protein is encoded in a sequence of precisely organized DNA 'letters'. Unlocking these instructions requires an enzyme known as RNA polymerase (RNAP for short), which reads the DNA segment and faithfully copies the information to form a strand of RNA. This molecule then relays the genetic message to the machinery that pieces together a protein.

An RNAP works by reading a DNA segment and building a matching RNA strand at the same time. The enzyme clamps onto DNA, and threads it letter-by-letter through its reading and building site. For each DNA letter that RNAP reads, the enzyme adds a matching RNA building block onto the budding RNA strand, with DNA and RNA segments then being moved away from the active site.

However, RNAP does not usually read a whole gene in one go: there are several 'pause sites' in the sequence where it stops and waits for instruction. If the cell needs this protein immediately, it sends signals that encourage RNAP to carry on and even ignore further pause sites; if the protein is not needed at the time, the enzyme is instructed to terminate the RNA-making process. This mechanism is present in species across the tree of life, and is key so that a cell fine-tunes its protein production.

Once RNAP has stopped, several well-studied mechanisms kick in to stabilize the enzyme in its waiting position. Yet, it is still unclear how the enzyme, which normally reads 50 to 100 DNA letters per second, is able to come to a halt in the first place.

To dissect this mechanism, Saba et al. made targeted changes to RNAP or to the DNA segment it was reading, and then closely monitored the movement of the protein under these conditions. The experiments revealed that when RNAP interacts with multiple signals in the DNA, such as particular sequences just before or inside the segment being read, the enzyme changes its structure slightly, and loosens its grip on DNA and RNA. With the enzyme's new shape, the RNA strand is ready to be extended, but the DNA segment is trapped and cannot move into the reading site. This prevents a new RNA letter to be added onto the growing strand, stopping RNAP in its tracks.

Knowing how RNAP pauses may help researchers to understand how its activity is regulated, for example by antibiotics. Ultimately, this could allow us to manipulate the activity of the enzyme so that we could control how and when a cell creates specific proteins.

DOI: https://doi.org/10.7554/eLife.40981.002

exit channel that alters RNAP conformation (at least in bacteria), or by interactions of diffusible regulators with the ePEC. Recent cryoEM structures of artificially assembled PECs suggest the ePEC is half-translocated with a tilted RNA–DNA hybrid, meaning that the RNA but not the DNA is translocated and the next template base is still sequestered in the downstream DNA duplex (*Kang et al., 2018a*; *Guo et al., 2018*; *Vos et al., 2018*). The hairpin-stabilized PEC is additionally inhibited by a rotation of the swivel module (including the clamp, shelf, and SI3) that inhibits trigger-loop folding (*Kang et al., 2018a*; *Guo et al., 2018*).

High-throughput sequencing of nascent RNAs from bacteria (NET-seq) reveals a consensus elemental pause sequence conserved among diverse bacterial RNAPs and mammalian RNAPII whose effects on pausing in vitro are consistent with a block in template DNA translocation sometimes accompanied by modest backtracking (*Larson et al., 2014*; *Vvedenskaya et al., 2014*; *Imashimizu et al., 2015*). Although earlier work establishes contributions of multiple pause signal components (upstream RNA, RNA–DNA hybrid, downstream fork junction, and downstream DNA) to hairpin-stabilized pausing (*Chan and Landick, 1993*; *Wang et al., 1995*; *Chan et al., 1997*), the definition of the consensus elemental pause signal has varied and it is unknown if the discrete components affect a common step in the elemental pause mechanism.

Additionally, questions remain about the structure and properties of the ePEC. A longstanding debate is whether the ePEC is an on-pathway state unable to translocate DNA or RNA in a largely unchanged RNAP due to the thermodynamic properties of the RNA–DNA scaffold (i.e., a pretranslocated pause; *Bai et al., 2004*; *Bochkareva et al., 2012*) or if it represents an offline state generated by conformational rearrangement of RNAP that forms in kinetic competition with the on-pathway steps (*Landick, 2006*; *Herbert et al., 2006*; *Kireeva and Kashlev, 2009*; *Imashimizu et al., 2013*;

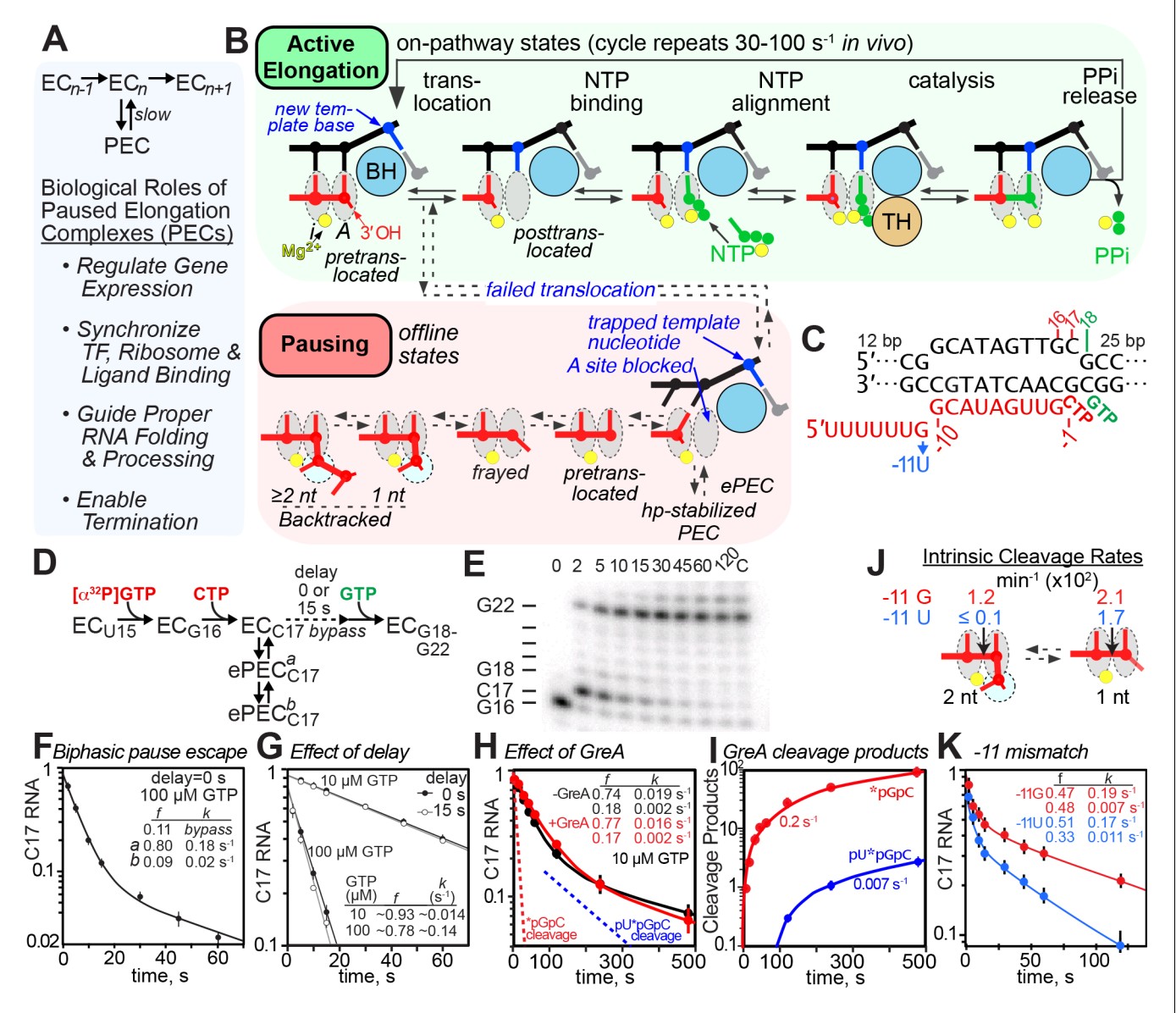

**Figure 1.** The consensus elemental pause signal. (A) The simplest elemental pause kinetic scheme and the biological roles of pausing. (B) Model of RNAP active site during the normal nucleotide addition cycle (*green*) consisting of RNA-DNA translocation, NTP substrate binding, active-site closing and NTP alignment (mediated by the trigger loop [TL] and bridge helix [BH]), phosphoryl transfer (catalysis), and PPi release. Catalysis is reversible when PPi remains present. Pausing (*red*) occurs when altered RNAP-DNA-RNA interactions create a barrier to completion of translocation that blocks entry of the next template nucleotide into the active site, creating the elemental pause state. The elemental pause state can rearrange further into backtracked or hairpin-stabilized pauses depending on other RNA-DNA sequences. (C) Central region of the RNA:DNA scaffold used for pause assays (complete scaffold is in *Figure 1—figure supplement 1A*). The −11U mismatch substitution was used to test possible contributions of backtracking (see panels J and K). (D) Pause assay and kinetic scheme consistent with biphasic pause escape intermediates. (E) Example of pause assay products separated by denaturing gel electrophoresis (radiolabeled by [32]P-G16 incorporation; C17 is the pause RNA). (F) Observed levels of C17 RNA as a function of time in a simple elemental pause assay at 37°C and 100 µM GTP. Pause fractions (*f*) and apparent escape rates (*k*) are shown for the pause and slow pause species. (G) Effect of delayed addition of 10 µM or 100 µM GTP after formation of C17 PECs. Additional data documenting the offline elemental pause states are in *Figure 1—figure supplement 1B–G* and *Figure 1—figure supplement 2*. (H) Effect of GreA (1 µM) on elemental pausing at 10 µM GTP. The dotted lines indicate the rates of 2-nt or 3-nt cleavage in the presence of GreA based on appearance of cleavage products (see panel I). (I) Rate of appearance of GreA-induced cleavage products during pause assay in panel H. (see *Figure 1—figure supplement 3A,B,C,D* for measurements of GreA/B cleavage rates and effects on pausing). (J) Rates of intrinsic cleavage of C17 halted ePECs matched with pause assays shown in panel (K). The −11U substitution reduced intrinsic cleavage in the −1 backtrack register by a factor of >50 (see *Figure 1—figure supplement 3E,F* for measurement of intrinsic cleavage rates). (K) Effect of −11 U mismatch on the elemental pause at 100 µM GTP with pause prone-RNAP.

DOI: https://doi.org/10.7554/eLife.40981.003

*Figure 1 continued on next page*

*Figure 1 continued*

The following figure supplements are available for figure 1:

**Figure supplement 1.** The elemental pause is a distinct offline state, not an on-pathway elongation intermediate.
DOI: https://doi.org/10.7554/eLife.40981.004

**Figure supplement 2.** (A) Scaffold used to probe for biphasic escape kinetics at two sequential elemental pause sites (Tandem Consensus Elemental Pause Scaffold; tDNA #12623, ntDNA #12624, RNA #8342; *Supplementary file 1*).
DOI: https://doi.org/10.7554/eLife.40981.005

**Figure supplement 3.** (A) Model for RNA cleavage in pre-translocated or backtracked states following entry into the elemental pause.
DOI: https://doi.org/10.7554/eLife.40981.006

*Kang et al., 2018a*). Uncertainty also exists as to whether the elemental pause is non-backtracked (*Landick, 2006*; *Herbert et al., 2006*; *Kireeva and Kashlev, 2009*; *Kang et al., 2018a*) or must be backtracked one or more registers (*Dangkulwanich et al., 2013*; *Forde et al., 2002*; *Galburt et al., 2007*; *Mejia et al., 2015*; *Tadigotla et al., 2006*; *Ó Maoiléidigh et al., 2011*).

To address these questions, we combined kinetic analyses of pausing using elemental pause sequence variants or mutant RNAPs with precise structural probes of translocation, trigger-loop folding, and clamp conformation. Our results lead us to propose a multistate model of elemental pausing in which template-base loading in a half-translocated offline intermediate limits pause escape.

## Results

### The ePEC is an offline state formed in a branched kinetic mechanism

To probe the elemental pause mechanism, we used kinetic analyses of ECs reconstituted on a synthetic RNA-DNA scaffold encoding a consensus elemental pause sequence (*Figure 1C* and *Figure 1—figure supplement 1A*; *Larson et al., 2014*). We first asked if the pause signal always causes ECs to bifurcate into paused and rapidly elongating (bypass) fractions. We measured C17 pause RNA as a function of time when radiolabeled G16 ECs were extended with 100 µM each CTP and GTP (*Figure 1D,E,F*). Most ECs (80% at 37°C) entered a paused state (lifetime 5 s; *a*, *Figure 1D,F*), whereas some ECs (20%) transcribed past C17 rapidly (lifetime 0.1 s; *bypass*, *Figure 1D,F*). Invariably, a minor ePEC population (typically 15%) escaped more slowly (lifetime 100 s; *b*, *Figure 1D,F*), requiring double-exponential fitting of the escape rate. Pause bypass was evident from y-intercepts <1 in the double-exponentials fits (*Figure 1F*). Interestingly, the amounts of slow ePEC and bypass fractions varied among RNAP preparations (*Figure 1—figure supplement 1B*). This kinetic malleability of ePECs differed from the reproducible kinetics typically observed for hairpin-stabilized pauses like the *his*PEC (*Toulokhonov et al., 2001*; *Kyzer et al., 2007*; *Kang et al., 2018a*).

To confirm that the ePEC is an offline state formed in competition with bypass nucleotide addition (i.e., in a branched kinetic mechanism; *Figure 1D*), we performed three additional tests. First, we asked whether the fraction of ECs entering the ePEC state increased if GTP (the NTP required for pause escape) were withheld to allow more time for ePEC formation. Unlike for the hairpin-stabilized *his*PEC (*Toulokhonov et al., 2007*), halting ECs at C17 for 15 s before GTP addition did not change the 10–20% EC that bypassed the pause (*Figure 1G*). However, the bypass fraction was shifted from ~22% at 100 µM GTP to ~7% at 10 µM GTP, suggesting that GTP can affect a partition between the on-pathway EC and offline elemental pause state. Second, using data from *Larson et al., 2014*, we found that the bypass fraction varied but appeared to plateau at high GTP (*Figure 1—figure supplement 1C*). Because extrapolation of bypass from bi-exponential fits is an approximation, we next used numerical integration of rate equations as a third test to ask if reaction progress curves are better fit by an unbranched or branched kinetic mechanism at saturating GTP (10 mM; *Figure 1—figure supplement 1A,F*; Materials and methods). The unbranched (on-pathway pause) mechanism did not fit the data well, thus favoring a branched mechanism (*Figure 1—figure supplement 1F*).

We also used kinetic modeling to reexamine the proposed on-pathway pausing reported by *Bochkareva et al., 2012*. Using their optimal pause scaffold sequence, we replicated the reported high-efficiency pausing (*Figure 1—figure supplement 1D,E*). However, this scaffold positions the downstream DNA end (+17) within RNAP, so that forward translocation reduces downstream DNA-

RNAP interaction. Consistent with a prior finding that downstream DNA truncation increases pausing (*Kyzer et al., 2007*), pause bypass was readily detected when the downstream DNA end was positioned outside RNAP as in natural ECs (*Figure 1—figure supplement 1D,E*). This result highlights why highly efficient pauses, even if they isomerize to an offline state, may appear to be on-pathway when bypass falls below ~5%. We confirmed the branched kinetic pause mechanism on the Bochkareva *et al.* scaffold at saturating GTP by kinetic modeling (*Figure 1—figure supplement 1G*). In agreement with most prior ensemble (*Kassavetis and Chamberlin, 1981*; *Kyzer et al., 2007*; *Toulokhonov et al., 2007*; *Strobel and Roberts, 2015*) and single-molecule (*Herbert et al., 2006*; *Larson et al., 2014*; *Gabizon et al., 2018*) analyses of pausing, we conclude that the elemental pause is an offline state that forms in competition with on-pathway nucleotide addition (i.e., in a branched kinetic mechanism).

## The consensus ePEC readily backtracks by one bp but reversal of one-bp backtracking does not limit the rate of pause escape

The existence of a minor, variable, slowly escaping ePEC population has been a consistent and puzzling feature of pause assays using synthetic scaffolds (*Figure 1F*). We wondered if the slow pause population could be explained by backtracking, an EC conformational change, or a subpopulation of chemically altered RNAP. Variation of the slow ePEC fraction among RNAP preparations (*Figure 1—figure supplement 1B*) seemed to favor a chemically altered subpopulation. We found that absence of neither the ω subunit nor the αCTDs changed the slow ePEC fraction (not shown). More definitively, transcription through tandem consensus pause sequences revealed formation of the slow fraction from all RNAPs arriving at the second pause site rather than a chemically altered, slow subpopulation of RNAP that would be filtered out by the first pause site (*Figure 1—figure supplement 2*).

We next used GreA- or GreB-induced cleavage to ask if the minor slow ePEC fraction was backtracked. Either or both GreA or GreB had little effect on pause lifetimes, even though 2-nt cleavage products, indicative of a 1 bp backtracked state, appeared much faster than the rate of ePEC escape and 3-nt cleavage products, indicative of a 2 bp backtracked state, were detectable (*Figure 1H,I*; *Figure 1—figure supplement 3B,D*). Thus, even though the scaffold structure disfavors $\geq$2 bp backtracking due to a –12 rU–dG mismatch, ePECs may shift from the 1 bp backtracked state to $\geq$2 bp backtracked states, at least in the presence of GreA, GreB, or both. We next tested the effect on both pausing and intrinsic cleavage of a −11rU-dC mismatch that would disfavor even 1 bp backtracking using an RNAP that formed a high level of slow fraction (*Figure 1C,K,J*; *Figure 1—figure supplement 3E,F*). The −11rU-dC mismatch virtually eliminated $\geq$2 nt cleavage indicative of backtracking, reduced 1-nt cleavage indicative of the pretranslocated register, and modestly decreased the slow pause fraction.

The relative ease with which ePECs entered backtracked registers is notable given the half-translocated state observed in ePEC cryo-EM structures (*Kang et al., 2018a*; *Guo et al., 2018*; *Vos et al., 2018*). Backtracking of ePECs is detectable both in vivo and in vitro (*Larson et al., 2014*; *Imashimizu et al., 2015*), but the half-translocated or pretranslocated states appears to be predominant (*Figure 1J*; *Larson et al., 2014*). Our GreA/B and intrinsic cleavage results establish that slow escape from a 1 bp backtrack state explains neither the major nor the minor pause fractions even though the 1 bp backtrack state is accessible. A parsimonious explanation is that the ePEC readily equilibrates among several active-site states, including half-translocated, pretranslocated, and backtracked, with the half-translocated state being the dominant species in which the kinetic block to pause escape is manifest. The slow pause state may be backtracked by $\geq$2 bp (since 0.007 s$^{-1}$ 3-nt cleavage is close to the 0.002 s$^{-1}$ escape rate; *Figure 1H,I*), but its persistence in the −11U mutant suggests other changes to the ePEC must also contribute (see Discussion).

## The elemental pause signal is multipartite

Available NET-seq data have been interpreted to suggest an elemental pause signal involving just the upstream fork-junction (usFJ) and downstream fork-junction (dsFJ) sequences or a multipartite signal that additionally depends on the hybrid (Hyb) and downstream DNA (dsDNA) sequences (*Larson et al., 2014*; *Vvedenskaya et al., 2014*; *Imashimizu et al., 2015*). To test the proposed roles of Hyb and dsDNA sequences and additionally to ask if the different pause signal elements

combine additively to define a single energetic barrier to pause escape, as observed for the hairpin-stabilized *his* pause signal (*Wang et al., 1995*; *Chan et al., 1997*), we analyzed effects on pausing of substitutions in each element separately and in combination (*Figure 2A,B*).

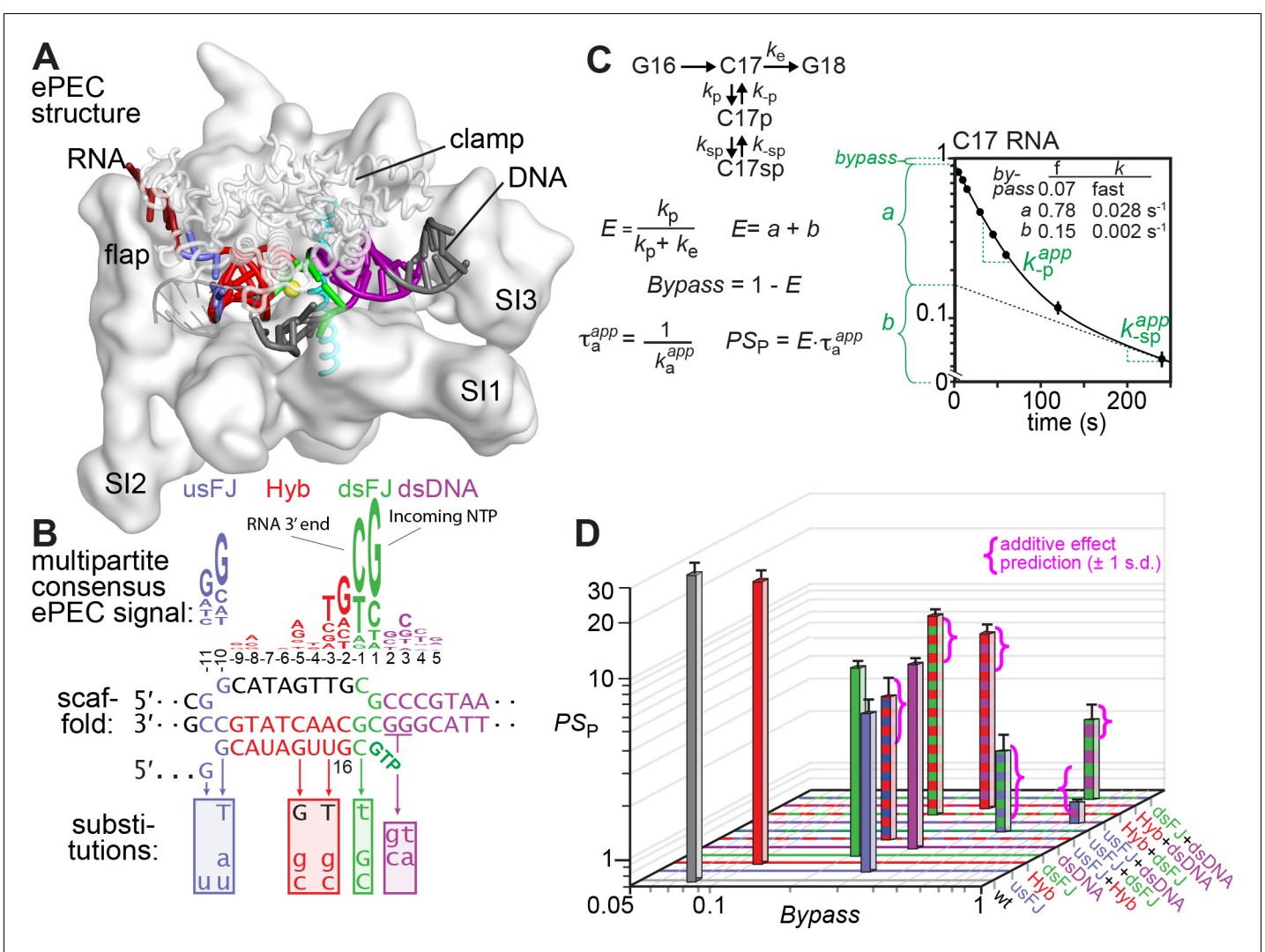

**Figure 2.** The elemental pause signal is multipartite. (A,B) The consensus ePEC. The ePEC structure (pdb 6bjs; *Kang et al., 2018a*) is shown above the consensus pause sequence (*Larson et al., 2014*) color-coded as usFJ (blue), Hyb (red), dsFJ (green), and dsDNA (purple). Scaffold substitutions tested for effects on pausing are shown below the location of the changes. (C) Two-component pause mechanism, equations used to calculate pause efficiencies and pause strengths, and example plot of pause assay data. The pause assay data shown are for the 'wild-type' (unsubstituted) consensus pause scaffold at 37°C and 10 μM GTP. Pause fractions (fast fraction *a*, slow fraction *b*) and escape rates $k_{\text{-p,app}}$ and $k_{\text{-sp,app}}$ were obtained by nonlinear regression using a double-exponential equation (see Materials and methods). (D) Plot of pause strength (PS$_P$) *vs.* bypass for each scaffold variant. Combinations are indicated by '+" (e.g., usFJ + Hyb). The magenta brackets show the predicted range of PS$_P$ values (95% confidence interval) for combinations of substitutions in individual pause signal elements assuming the pause signal elements additively and independently affect a single energy barrier to pause escape (additive effects on ΔG$^‡$ are multiplicative in PS$_P$). Additional data for the wild-type and mutant pause signals are in *Figure 2—figure supplement 1*.

DOI: https://doi.org/10.7554/eLife.40981.007

The following source data and figure supplement are available for figure 2:

**Source data 1.** Fraction C17 pause RNA for wild-type and mutant pause signals.
DOI: https://doi.org/10.7554/eLife.40981.009
**Figure supplement 1.** (A) Quantitation of pause assays with scaffold variants described in *Figure 2*.
DOI: https://doi.org/10.7554/eLife.40981.008

We measured pause lifetimes and the bypass, pause, and slow pause fractions using two-exponential fits of C17 RNA *vs.* time at 10 µM GTP (*Figure 2C*; *Figure 2—figure supplement 1A*). Because a translocation barrier would affect both pause formation and escape, we plotted pause strength (PS; pause efficiency times the lifetime [$\tau$] of the major pause species) vs. pause bypass fraction (*Figure 2D*). Alone, the usFJ, dsFJ, and dsDNA mutants reduced PS by a factor of ~5, whereas the Hyb mutant decreased PS by a factor or ~1.4 fold (but see larger effects in *Bochkareva et al., 2012*). Combinations of mutants produced additive effects on pause strength (*Figure 2D*, magenta brackets; additive effects on an energy barrier are multiplicative in $\tau$ or PS; e.g., reduction of PS by factors of 1.4 and 3.7 for the Hyb and dsDNA substitutions predicts a combined reduction by a factor of 5.1 *vs.* the factor of 5.2 observed). Additive effects are expected if pause signal components independently affect the same energetic barrier to pause escape (e.g., translocation of the template base).

The lifetimes of the major and minor pause states were highly correlated for the pause-sequence variants (*Figure 2—figure supplement 1B*). In contrast, the slow pause fraction ($E_b$; *Figure 2—figure supplement 1C*) was uncorrelated with the major pause escape rate ($k_{sp}^{app}$), the major pause fraction ($E_a$), and the total ePEC fraction ($E$). These results are consistent with a model in which both the major pause species and the minor, slowly escaping pause species pass through a common barrier for pause escape (e.g., template-base loading). The significant variation in the amount of slow pause species provides additional evidence that the slow species arises by kinetic partitioning of a single RNAP population and not from a chemically distinct 'slow' subpopulation of RNAP.

We conclude that the elemental pause signal is indeed multipartite with significant contributions from sequences in the usFJ, Hyb, dsFJ, and dsDNA to a common kinetic barrier to escape from multiple paused states.

## Hybrid translocation is not rate-limiting for elemental pause escape

To ask if the barrier to elemental pause escape corresponds to the half-translocated intermediate identified by cryoEM (*Kang et al., 2018a*; *Guo et al., 2018*), we next tested for ePEC translocation at the usFJ and dsFJ using a fluorescence-based translocation assay developed by Belogurov and co-workers (*Figures 3A* and *4A*; *Malinen et al., 2012*). In this assay, the fluorescent guanine analog 6-methylisoxanthopterin (6-MI) is located in the template DNA (usFJ) or nontemplate DNA (dsFJ) such that an adjacent guanine unstacks from 6-MI upon translocation of the hybrid or the dsDNA; this unstacking increases 6-MI fluorescence (*Figure 3A*). We compared translocation of the hybrid or the dsDNA leading to increased 6-MI fluorescence on ePEC scaffolds to a control scaffold previously found to translocate rapidly upon 3′-CMP addition (*Figure 3B*; *Malinen et al., 2012*; *Hein et al., 2014*). Because both the control and ePEC scaffold encode C as the RNA 3′ nucleotide and G as the next nucleotide after translocation (*Figures 3B,C and* and *4A,B*), differences in their behavior can be attributed to the usFJ, Hyb, and dsDNA. To calibrate the 6-MI signal change from the posttranslocated hybrid, we compared the effects of CMP, 3′dCMP, or 2′dCMP (*Figure 3D–F*). A 3′deoxy ribonucleotide shifts the hybrid toward the pretranslocated register, whereas a 2′deoxy ribonucleotide shifts the hybrid toward the posttranslocated register (*Malinen et al., 2012*). For both scaffolds, 2′dCMP gave a 6-MI signal increase comparable to CMP, whereas 3′dCMP reduced the signal increase. These results suggested that addition of CMP shifts both hybrids toward the posttranslocated register, although the absolute fluorescence change for the ePEC scaffold was about half that of the control scaffold. Rapid-quench and stopped-flow kinetic measurements revealed that CMP added rapidly (~300 s$^{-1}$) followed by a rapid (~200 s$^{-1}$) rise in fluorescence signal indicating translocation on both scaffolds. In contrast, subsequent extension to G18 of the ePEC but not the control EC was slow (~0.2 s$^{-1}$ *vs.* 40 s$^{-1}$; *Figure 3D,E*).

Because the 6-MI and −10 substitutions in the ePEC scaffold could affect pausing, we verified that the 6-MI ePEC scaffold produced pause fractions (~85%) and lifetimes similar to the unmodified consensus scaffold (*Figure 3G,H*). Thus, the ePEC fluorescence signal could not be explained by the ~15% bypass fraction and must have arisen in significant part from the paused species. We also verified that CMP addition occurred fully in the fluorescence assay (*Figure 3—figure supplement 2*). We conclude that the ePEC hybrid translocates rapidly after CMP addition, resembling the control scaffold. The reduced level of unquenching (~50% of the control scaffold) could reflect lower 6-MI fluorescence in a half-translocated hybrid, since the ePEC cryoEM structure suggests −10 dC (*vs.* −10 dG in the assay scaffold) remains at least partially paired in the hybrid with an altered interaction

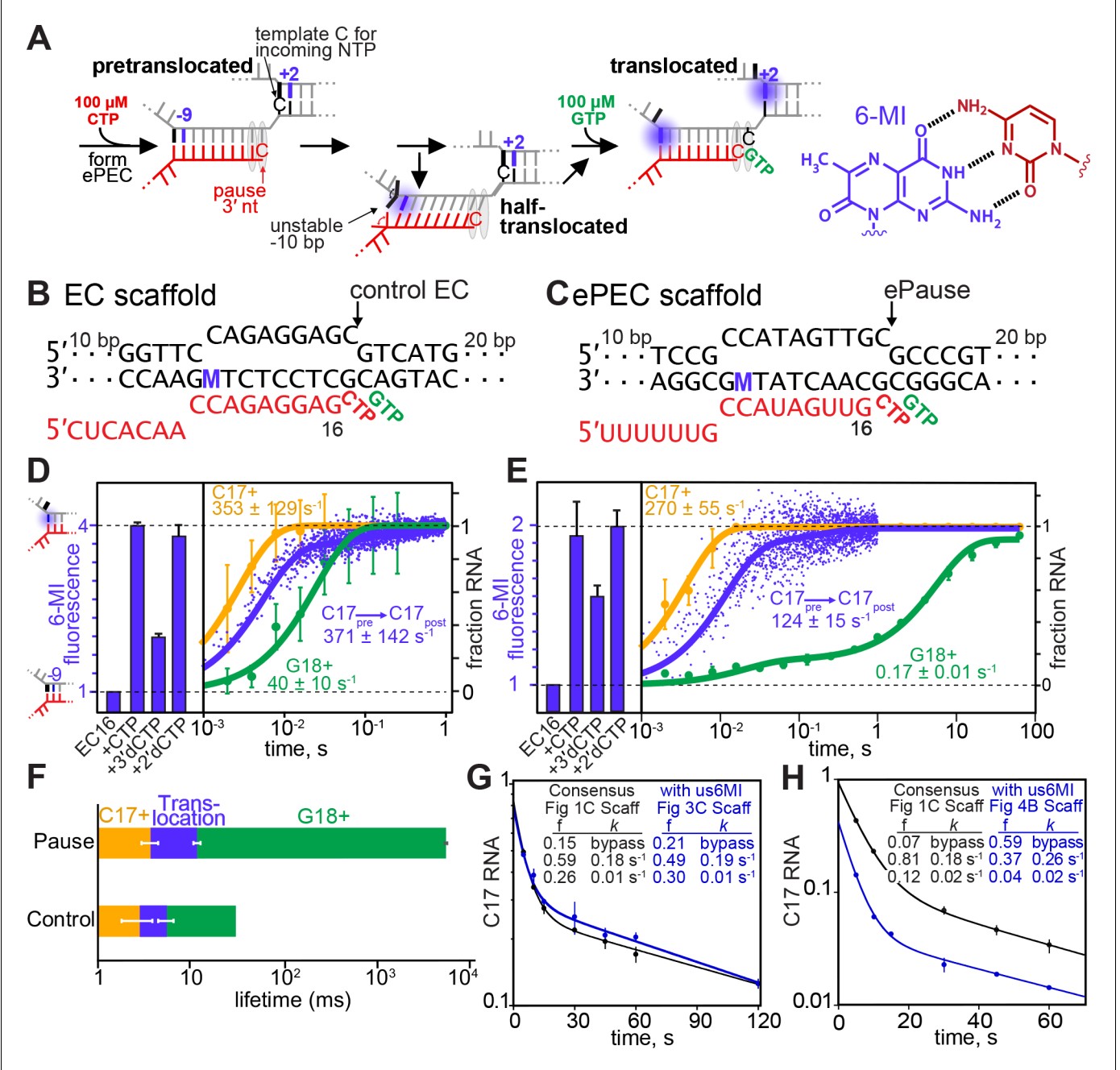

**Figure 3.** Translocation of the RNA:DNA hybrid is not rate-limiting for elemental pause escape. (A) Scheme for translocation following CTP addition to control EC or ePEC scaffolds. The locations of 6-MI for both usFJ and dsFJ probes are indicated, but a probe was present in only one location in the scaffolds used (see panels B and C and *Figure 4A and B*). 6-MI is a GMP base analog that fluoresces unless quenched by stacking with an adjacent GMP. (B and C) Scaffolds used to reconstitute control EC and ePEC for fluorescence experiments. M, position of 6-MI. Full scaffold sequences and additional data characterizing pause behaviors of these scaffolds are in *Figure 3—figure supplement 1* and *Figure 3—figure supplement 2*. (D) Equilibrium fluorescence changes (blue bars) upon addition of incoming NTP or NTP analog to the non-pausing EC scaffold measured at 37°C. Nucleotide addition (orange and green traces) and translocation (blue trace) rates were determined using KinTek quenched-flow (RQF-3) or stopped-flow (SF-300X) instruments, respectively. C17+ represents the fraction of RNA at or beyond C17. G18+ represents the fraction of RNA at or beyond G18. 6-MI fluorescence was normalized to that observed after 2'dCMP addition to G16. (E) Experiments performed as in D but on an ePEC scaffold. (F) Data from (D) and (E). Mean times for CMP addition, translocation, and subsequent GMP addition for the ePEC and control EC. (G,H) Pause kinetics at 100 μM GTP for the 6-MI-containing ePEC scaffolds (G), us6MI; (H), ds6MI) compared to the consensus pause scaffold. f, kinetic fractions. k, rate constants for pause fractions.

DOI: https://doi.org/10.7554/eLife.40981.010

*Figure 3 continued on next page*

*Figure 3 continued*

The following figure supplements are available for figure 3:

**Figure supplement 1.** Control EC scaffold used for 6-MI translocation assays in *Figures 3* and *4*.

DOI: https://doi.org/10.7554/eLife.40981.011

**Figure supplement 2.** Reconstitution of ECs and PECs for 6-MI translocation assays.

DOI: https://doi.org/10.7554/eLife.40981.012

with the lid (*Kang et al., 2018a*). Alternatively, –10 dG may unstack in our assay at 37°C but the pre-translocated and half-translocated hybrids may be in dynamic equilibrium (consistent with intrinsic hydrolysis readout; *Figure 1J*). These results also confirm that most ePECs are not backtracked, which would not unquench 6-MI (*Figure 1I* and *Figure 1—figure supplement 3*). Together the data are fully consistent with a view that the ePEC, once formed, may easily fluctuate among multiple translocation states.

## Downstream DNA translocation is rate-limiting for elemental pause escape

To test whether dsDNA fails to translocate in the ePEC (as predicted from the ePEC structure), we used an ePEC scaffold for which translocation of the +1 dC–dG bp would generate a +2 6-MI fluorescence signal (+1 dG would shift into an RNAP pocket that aids pause escape; *Vvedenskaya et al., 2014*). Placement of a dT–dA bp at +3 was necessary to generate a strong +2 6-MI signal. The changes needed for the 6-MI assay weakened the elemental pause signal but still allowed ECs to enter the pause state (*Figure 3H*), possibly to greater extent prior to GTP addition (*e.g.*, see effect of GTP in *Figure 1G*). For the control scaffold, addition of CMP or 2′dCMP or extension with CTP + GTP to G19 gave a large 6-MI signal indicating translocation after C17 nucleotide addition (*Figure 4C*). In contrast, the ePEC scaffold gave minimal 6-MI signal after CMP or 2′dCMP addition (consistent with translocation in only a bypass fraction) compared to extension to G19, which produced a strong 6-MI signal (*Figure 4D*). These data confirm that the dsDNA in the ePEC

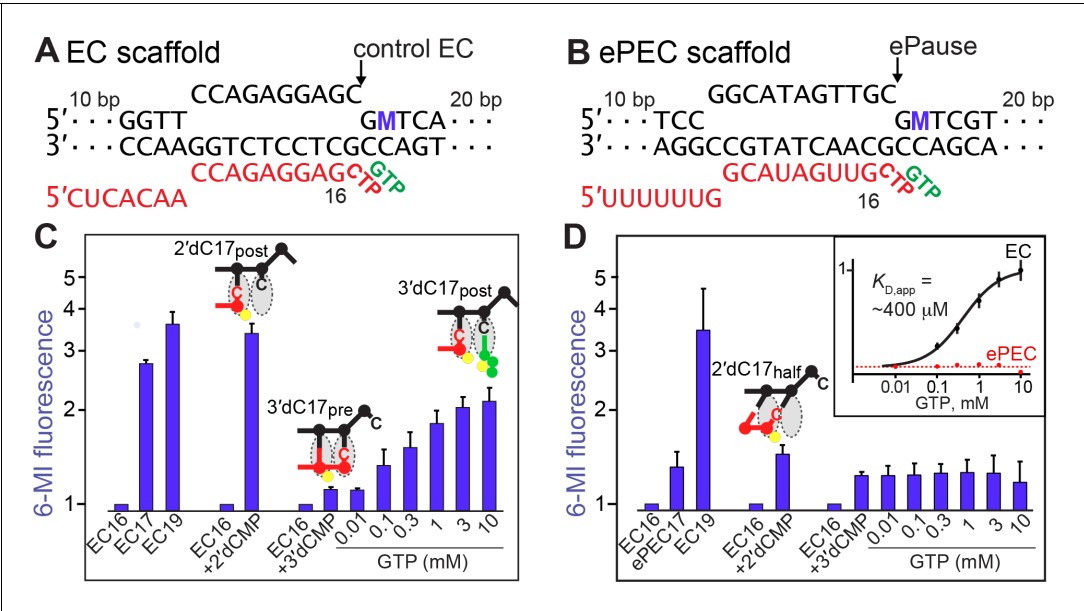

**Figure 4.** Translocation of the incoming template DNA limits elemental pause escape. (**A and B**) Scaffolds used to reconstitute control EC and ePEC for fluorescence experiments. M, position of 6-MI. (**C and D**) Equilibrium fluorescence changes of control EC and ePEC, respectively, upon reaction with NTP or NTP analog or binding of GTP to 3′deoxyC-containing complexes. RNA extension was verified by quantifying radiolabeled RNAs (see *Figure 3—figure supplement 2*). Inset, GTP-binding curve fit for the 3′dC17 EC (black line) and the 3′dC-ePEC (dotted red line; no binding evident). Error bars are SD of triplicate measurements.

DOI: https://doi.org/10.7554/eLife.40981.013

does not translocate. Since the dsDNA in the ePEC did not translocate, we used GTP binding to 3′dCMP-ePEC and control 3′dCMP-EC to assess translocation propensity. For the control EC, a clear increase in 6-MI signal was evident as GTP bound in the active site with apparent $K_d \approx$ 400 μM (*Figure 4D*, inset). This signal, which was reduced by interference from the high levels of GTP present, indicated that the post-translocated register was stabilized by GTP binding to the control EC (*Figure 4C*; *Malinen et al., 2014*). However, even a high GTP concentration was unable to shift the 3′dC17 ePEC to the post-translocated register (*Figure 4D*). We conclude that even the weakened elemental pause signal in the dsFJ ePEC assay scaffold prevented dsDNA translocation and thus template-base loading. Taken together, the usFJ and dsFJ 6-MI data confirm that the half-translocated state detected in ePEC cryoEM structure also forms in actively transcribing complexes, and that template-base loading is the relevant barrier to nucleotide addition in the ePEC.

## NTP binding rather than TL folding limits escape from the elemental pause

As a second approach to interrogate the status of the RNAP active site in the ePEC, we tested the effect of the consensus pause sequence on formation of disulfide bonds between Cys substitutions engineered to report TL conformation (Cys-pair reporters, CPRs; *Nayak et al., 2013*). Previous studies established that a β′ 937 Cys substitution in the trigger helix near the active site crosslinks efficiently to a Cys substitution at β′ 736 when the trigger helices formed (folded CPR, F937-736; *Figure 5A*). Other CPRs (P937-687, U937-1135, or U937-1139) report the partially folded (P) or unfolded (U) conformations of the TL (*Figure 5A*). The CPRs are sensitive to TL conformation when oxidized by cystamine (CSSC) because the CPR disulfide competes with formation of mixed disulfides (*Figure 5B*). These CPRs combined with other probes and a cryoEM structure revealed that the hairpin-stabilized *his*PEC fails to add the next nucleotide because it forms a swiveled PEC conformation that inhibits TL folding (*Hein et al., 2014*; *Nayak et al., 2013*; *Kang et al., 2018a*). In contrast, the ePEC, whose cryoEM structure is not swiveled, did not exhibit constraints on TL conformation (compare to control EC; *Figure 5C,D,E*). If anything, the ePEC accessed the folded TH conformation more readily than did the control EC, consistent with ePEC access to the pretranslocated register (*Figure 1J*) that is thought to increase TL folding (*Malinen et al., 2014*; *Liu et al., 2016*). Thus, inhibition of TL folding does not appear to limit escape from the elemental pause.

Incorporation of a 3′-dNMP in ECs and PECs allows the CPRs to detect TL folding stimulated by NTP binding in the EC and its inhibition in the *his*PEC (*Nayak et al., 2013*). Consistent with results of the 6-MI dsFJ translocation assay, F937-736 and P937-687 crosslinking in 3′dCMP-ePEC, and therefore TL conformation, were unaffected by high GTP concentration even though the crosslinks formed readily. In contrast, CPR crosslinking in control 3′dCMP-EC exhibited an obvious shift toward TL folding upon ATP binding (*Figure 5E,F*; *Nayak et al., 2013*). These results are consistent with the ePEC cryoEM structure and translocation assay results, supporting a view that inability to load the template base inhibits NTP binding in the ePEC.

## Clamp loosening but not extensive clamp opening accompanies elemental pausing

The role of clamp conformation in elemental pausing is uncertain. Although a crystal structure of *Tth*RNAP on a partial ePEC scaffold suggested the clamp could open in the ePEC (*Weixlbaumer et al., 2013*), more recent cryoEM structures and biochemical probing suggested the clamp remains closed in the ePEC but can swivel upon pause hairpin formation (*Guo et al., 2018*; *Kang et al., 2018a*). To probe the role of clamp conformation in elemental pausing, we examined the effect of stabilizing the clamp in the closed (unswiveled) conformation using an engineered disulfide bond between the lid and flap (β′258i-β1044; *Figure 6A and B*; *Kang et al., 2018a*). In contrast to suppression of pausing for the hairpin-stabilized *his*PEC (*Figure 6C*; *Hein et al., 2014*; *Kang et al., 2018a*), the closed-clamp disulfide had minimal effect on the ePEC lifetime (*Figure 6D*). We conclude that, unlike for hairpin-stabilized pausing and consistent with the cryoEM structure, clamp swiveling (or opening) is not required in the ePEC.

Even though full clamp swiveling is not required for elemental pausing, we wondered if any change in clamp conformation accompanied formation of the ePEC. To investigate this question, we used a variant of the disulfide bond probing strategy in which three Cys residues were positioned in

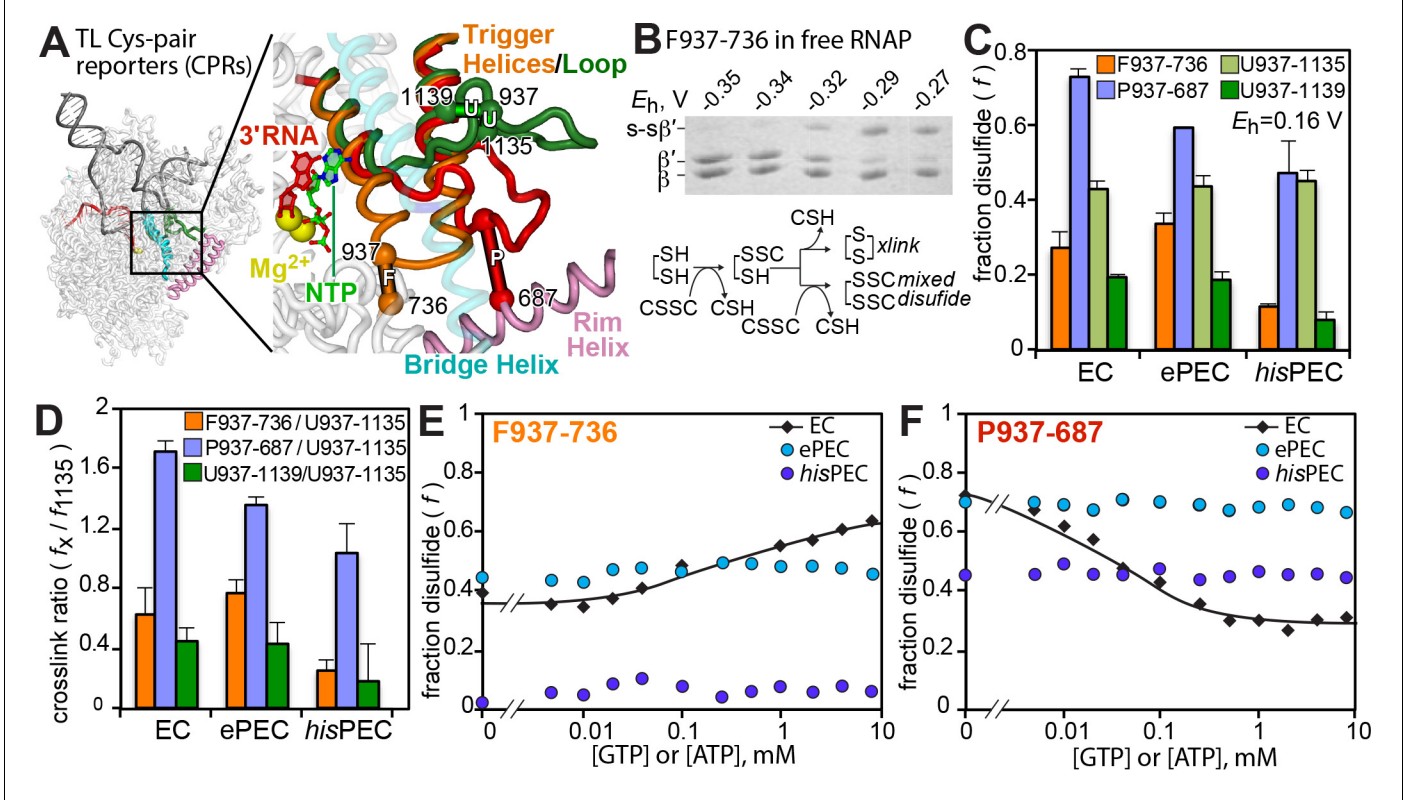

**Figure 5.** NTP binding but not TL folding is inhibited in the ePEC. Scaffolds used for these experiments are shown in *Figure 5—figure supplement 1*. (A) Location of F937-736, P937-687, U937-1137, and U937-1139 Cys-pair reporters to test various conformations of the TL. The SI3 domain inserted in the *Eco*RNAP TL is not shown. A complete description of these reporters is published elsewhere (*Nayak et al., 2013*; *Windgassen et al., 2014*). (B) Example CPR reaction showing SDS-PAGE mobility of uncrosslinked and crosslinked β′ as a function of redox potential generated by increasing concentrations of cystamine. Reaction scheme for crosslink generation by cystamine. (C) Fraction crosslink formed by CPRs for the EC, ePEC, and hairpin-stabilized *his*PEC in which TL mobility is restricted. Data are mean ± range from two replicates. (D) Ratio of crosslinks to the U1135 crosslink. A high ratio indicates a greater degree of crosslinking (E) F937-736 crosslink as a function of GTP or ATP concentration in 3′dC- or 3′dU-containing EC, ePEC, and *his*PEC. (F) P937-687 crosslink as a function of GTP or ATP concentration in 3′dC- or 3′dU-containing EC, ePEC, and *his*PEC.
DOI: https://doi.org/10.7554/eLife.40981.014

The following figure supplement is available for figure 5:

**Figure supplement 1.** Scaffolds used for disulfide bond assays of trigger loop position shown in *Figure 5*.
DOI: https://doi.org/10.7554/eLife.40981.015

RNAP such that either the closed-clamp disulfide (β′258i-β1044) or the swiveled-clamp disulfide (β′258i-β843) could form (*Kang et al., 2018a*). This Cys-triplet reporter (CTR) enabled a convenient measure of the energetic balance between closed and swiveled clamp conformations because the different crosslinked β′-β polypeptides could be readily distinguished by denaturing gel electrophoresis (*Figure 6B*; *Kang et al., 2018a*). The ratio of closed-to-swiveled crosslinks shifts during oxidation with cystamine likely because the mixed disulfide intermediates destabilize the closed-clamp conformation (*Figure 6E*). Thus, cystamine oxidation shifts the unswiveled-to-swiveled equilibrium toward the swiveled conformation, making it a sensitive assay of clamp conformation. This shift is greater for the *his*PEC (which favors clamp swiveling; *Kang et al., 2018a*), but intermediate between the EC and *his*PEC for the ePEC (*Figure 6F*). We conclude that formation of the ePEC is accompanied by a loosening of clamp contacts.

## A conserved arg in fork loop two may help inhibit DNA translocation at an elemental pause

We wondered if specific amino acids in RNAP inhibit template base translocation in the ePEC. Two candidates were β′K334 and βR542 (*Figure 7A*). β′K334 in switch two contacts the template DNA

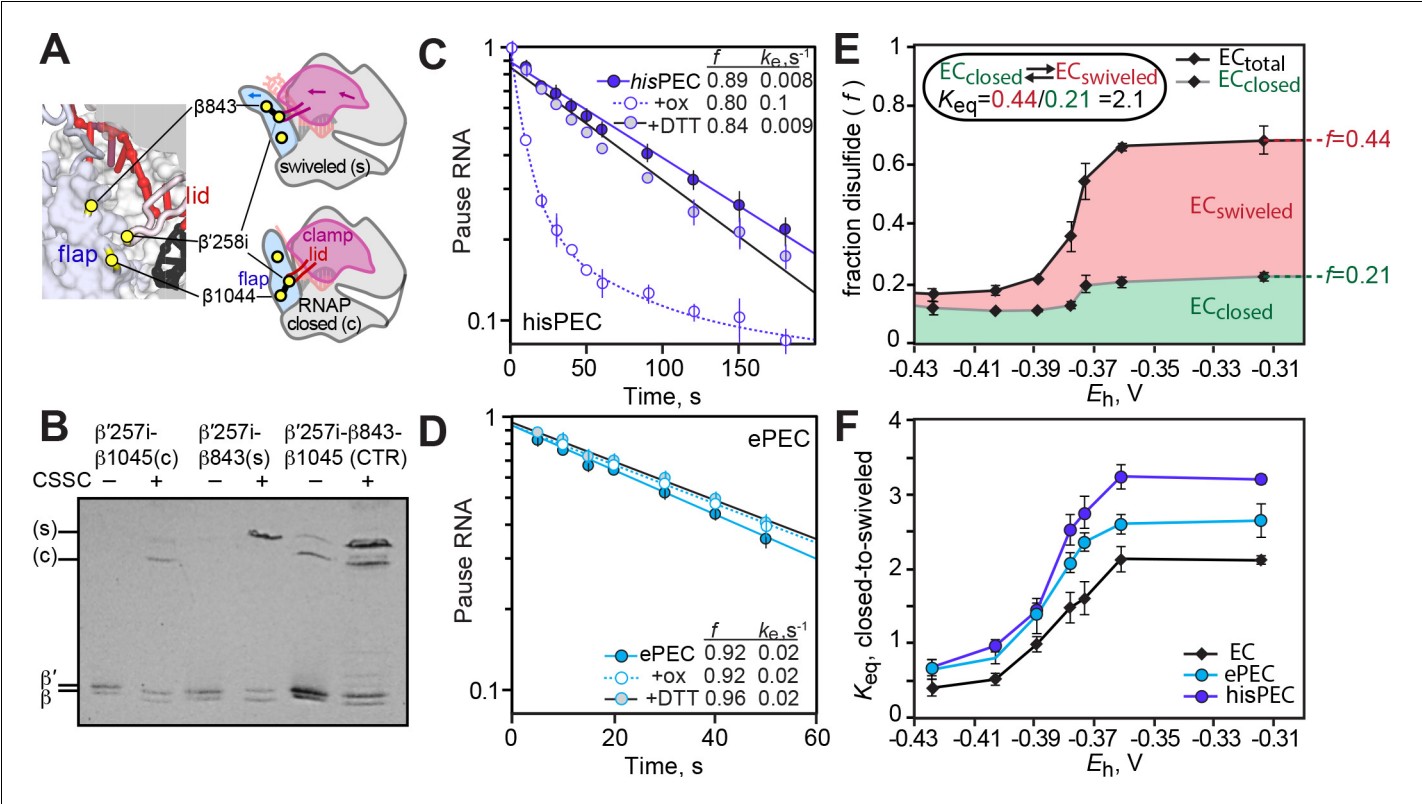

**Figure 6.** Restriction of clamp movement has less effect on ePEC than on hairpin-stabilized PEC. (**A**) Location of disulfides used to restrict clamp movement or generate the Cys-triplet reporter (CTR; described in *Hein et al., 2014*; *Kang et al., 2018b*). (**B**) Example β-β′ disulfide mobilities during SDS-PAGE illustrating the identification of the closed (β1044-β′258i, (**c**) or swiveled (β843-β′258i, (**s**) disulfides. (**C**) Effect of the closed clamp crosslink on pausing by the hairpin-stabilized *his*PEC (scaffold shown in *Figure 6—figure supplement 1A*). (**D**) Effect of the closed clamp crosslink on pausing by the ePEC (scaffold shown in *Figure 1—figure supplement 1A*). (**E**) CTR assay on an EC scaffold illustrating the formation of the closed and swiveled crosslinks as a function of redox potential generated by increasing concentrations of cystamine (*Figure 6—figure supplement 1F*). (**F**) CTR assay of clamp swiveling in EC, ePEC, and *his*PEC (scaffolds shown in *Figure 6—figure supplement 1B,C,D*, respectively).
DOI: https://doi.org/10.7554/eLife.40981.016

The following figure supplement is available for figure 6:

**Figure supplement 1.** Scaffolds used for CPR and CTR clamp-position assays shown in *Figure 6*.
DOI: https://doi.org/10.7554/eLife.40981.017

backbone adjacent to a dC blocked from active-site loading in half-translocated *Sce*RNAPII EC and *Tth*EC crystal structures (*Brueckner and Cramer, 2008*; *Weixlbaumer et al., 2013*) and the ePEC cryoEM structure. βR542 in fork loop two appears to contact the +1dC–dG bp in the ePEC cryoEM structure. Although the modest resolution of the cryoEM structure makes this assignment tentative and apparent H-bonding patterns differ, interaction of this conserved Arg with a pretranslocated template +1C also is seen in the half-translocated *Tth*RNAP crystal structure (*Weixlbaumer et al., 2013*), in a crystal structure of a *Tth*RNAP open complex formed on the *B. subtilis pyrG* promoter (*Murakami et al., 2017*), and in a 1 bp backtracked *Tth*RNAP PEC (*Sekine et al., 2015*). To ask if either β′K334 or βR542 played a key role in inhibiting template-base loading in the ePEC, we generated Ala substitution mutants and compared their pausing behaviors to wild-type RNAP. Because the strong consensus pause sequence could mask the effect of a single amino-acid contact, we also assayed the Ala mutants on the usFJ and dsFJ mutant templates that display weaker pausing behavior (*Figure 2B*). Strikingly, β′K334A resembled the wild-type enzyme on both consensus and mutant pause scaffolds, whereas βR542A decreased pausing by a factor of 2 on the consensus pause and dsFJ templates, but significantly more on the usFJ template (13X effect of usFJ *vs*. 6X for wild-type RNAP; *Figure 7B*). Given that βR542 interacts with the dsFJ, these data suggest that βR542

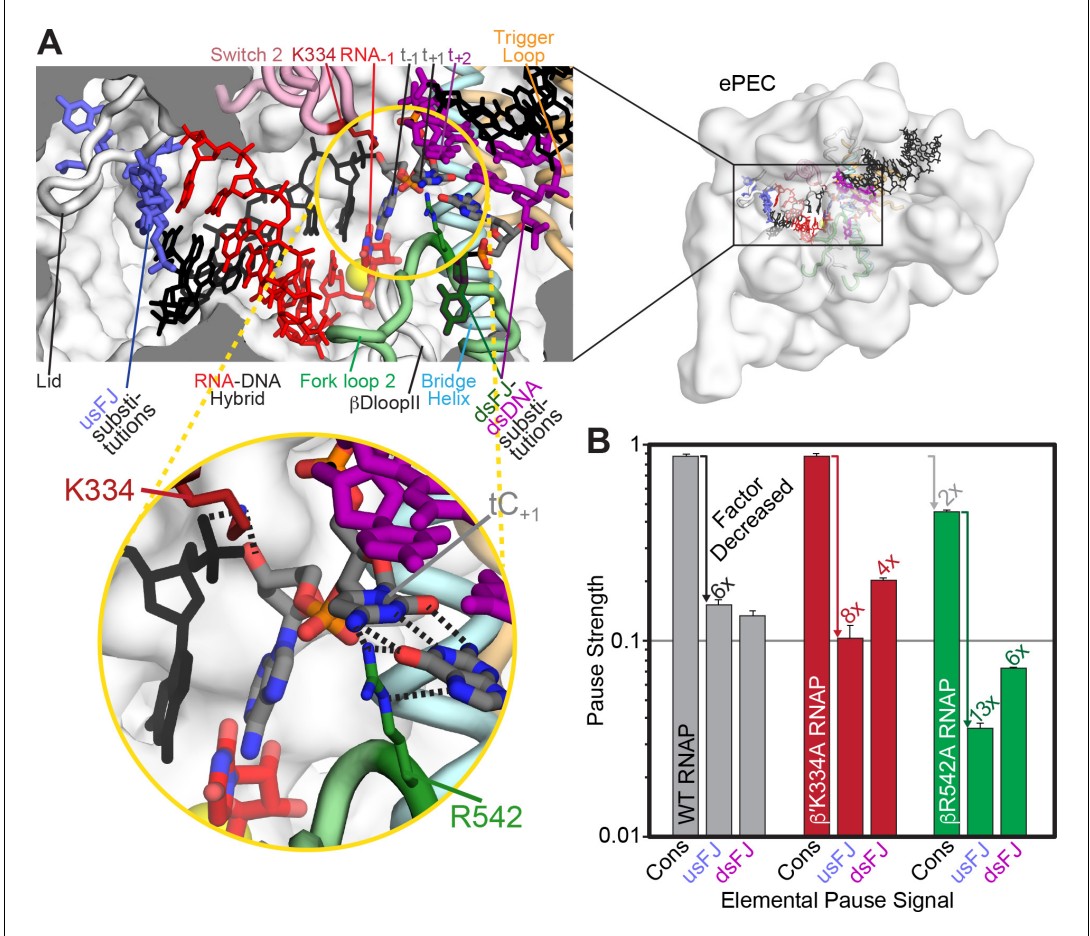

**Figure 7.** β R542 in fork-loop two may contribute to a template base loading barrier in ePEC. (**A**) Locations of β'K334 and βR542 in the ePEC. Relevant components of the ePEC are colored and labeled in a cutaway view of the active-site region of the ePEC. (**B**) Relative pause strength (PS$_P$, *Figure 2B*) for wild type, β'K334A, and βR542A RNAPs on the consensus pause scaffold (Cons), the usFJ mutant scaffold, and the dsFJ mutant scaffold (see *Figure 2A*). The factor decrease relative to the wild-type RNAP on the consensus pause scaffold or relative to the consensus pause for the different RNAPs is given above the bars and depicted using colored arrows. Error is SD from ≥3 replicates.

DOI: https://doi.org/10.7554/eLife.40981.018

The following source data is available for figure 7:

**Source data 1.** Fraction C17 pause RNA for wild-type and mutant pause signals.
DOI: https://doi.org/10.7554/eLife.40981.019

contributes to elemental pausing but less on a mutant template altered near its contact. We conclude that βR542 may help inhibit template dC loading in the ePEC.

## Discussion

Our mechanistic study of elemental pausing documents key features of this fundamental regulatory behavior of RNAP. The elemental pause is an offline state that forms in competition with rapid elongation; thus, the pause mechanism requires distinct EC and pause conformations rather than a single state with an energetic barrier to translocation. The elemental pause signal is multipartite. RNAP interactions with the usFJ, Hyb, dsFJ, and dsDNA contribute to the key barrier to pause escape: template-base loading from a half-translocated state that involves clamp loosening and βR542 interactions. Other translocation states that can prolong ePEC lifetime form readily. Based on these results, we will describe a model for ePEC formation and escape and discuss its implications for gene regulation.

## A model for elemental pause formation and escape

To explain how RNAP responds to an elemental pause signal, we propose a multi-state pause mechanism in which pause escape is principally inhibited by inability to load the template base into the active site of RNAP in a half-translocated, off-line intermediate (*Figure 8A,B*). When RNAP encounters an elemental pause signal, a modest shift in the mobile modules of RNAP that are in contact with RNA and DNA (the clamp, shelf, lid, rudder, switch regions, lobe, protrusion, fork loop 2, and βDloopII) occurs during translocation. This shift creates or reinforces an energetic barrier to completion of translocation from a half-translocated state in which the RNA transcript has translocated but the DNA template has not, corresponding to the tilted hybrid intermediate observed in cryo-EM structures (*Kang et al., 2018a*; *Guo et al., 2018*; *Vos et al., 2018*). Comparison of ePEC and post-translocated EC cryoEM structures reveals movements that slightly reposition these key mobile modules (compare green EC to magenta ePEC positions, *Figure 8A*). Key contacts preventing DNA translocation involve the lid, rudder, and switch 2 (*Kang et al., 2018a*; *Guo et al., 2018*) as well as apparent H-bonds of R542 in fork loop two to the +1dC–dG bp that may hinder its translocation into the active site (*Figures 7A* and *8*; compare to the conserved Arg in a *Tth*EC, which contacts the backbone phosphate of a template nucleotide loaded into the active site; *Vassylyev et al., 2007*). Notably the apparent R542 H-bonding contacts to the +1 dC–dG bp as well as to an unpaired but pretranslocated +1 dC in an initiation complex (*Murakami et al., 2017*) are not feasible for other +1 bases, possibly suggesting multiple ways R542 could inhibit template DNA translocation in an ePEC. The fraction of EC that partitions into the ePEC state and the height of the energetic barrier to completion of translocation and pause escape are both functions of the specific sequences present in the elemental pause signal (i.e., a suboptimal signal will capture fewer ECs for a shorter overall dwell time).

DNA and RNA in the ePEC can move backwards to form pretranslocated, frayed, 1 bp backtracked, and, much more slowly, ≥2 bp backtracked states (*Figure 8B*). At least at the consensus pause, the half-translocated, pretranslocated, and 1 bp backtracked states equilibrate but these equilibria are biased toward the half-translocated state. This bias explains why we observed a fast translocation signal for the RNA:DNA hybrid (*Figure 3*) and why the half-translocated intermediate forms in reconstituted ePECs analyzed by cryo-EM (*Kang et al., 2018a*; *Guo et al., 2018*; *Vos et al., 2018*). The fast equilibria explain why GreA rapidly cleaves a 1 bp backtracked ePEC state (*Figure 1H and I*; *Figure 1—figure supplement 3*); the 1 bp backtracked state quickly repopulates after cleavage even though it is only a small fraction of the total ePEC states. It is possible that GreA shifts the bias toward the 1 bp backtracked state, but mass action 'pulling' of ePECs into the 1 bp backtracked state is a sufficient explanation. Because the cleaved ePEC re-encounters a strong pause signal, we observed little effect of GreA cleavage on pause lifetime. Our Gre A/B cleavage data also suggest a much slower entry into ≥2 bp backtracked states, which may be related to the minor but variable slow fraction of ePECs (see below).

This multistate model of the elemental pause predicts that the pretranslocated and 1 bp backtracked states can contribute significantly to pause lifetime at some pause sequences despite being in rapid equilibrium with the half-translocated intermediate in which the kinetic barrier to pause escape (template-base loading) is manifest. If an ePEC spends only 50% of the time half-translocated and 50% pre-translocated or backtracked, then the pause dwell time will increase by a factor of ~2 relative to an ePEC that spends >95% of the time half-translocated. This is true despite the rapid equilibria because only the half-translocated intermediate can escape the pause and the other states diminish its effective concentration. Shifting to 25% or 10% half-translocated would lengthen the pause by factors of ~3 or~9, respectively, for the same reason. Thus, an elemental pause signal that favors the 1 bp backtracked state, although different from the consensus signal we studied, would increase pause lifetime. These behaviors have been observed in vitro directly (*Gabizon et al., 2018*) or by biasing ePECs using opposing force (*Galburt et al., 2007*; *Dangkulwanich et al., 2013*), as well as in vivo (*Imashimizu et al., 2015*). Further, multiple locations of the 1 bp backtracked RNA 3′ nucleotide have been observed or proposed (*Figure 8B*; *Sekine et al., 2015*; *Wang et al., 2009*; *Sosunova et al., 2013*; *Turtola et al., 2018*). From a mechanistic standpoint, possible entry into multiple 1 bp backtracked states means that their contribution to pause dwell times may be both increased and highly variable as a function of sequence. The multistate elemental pause model

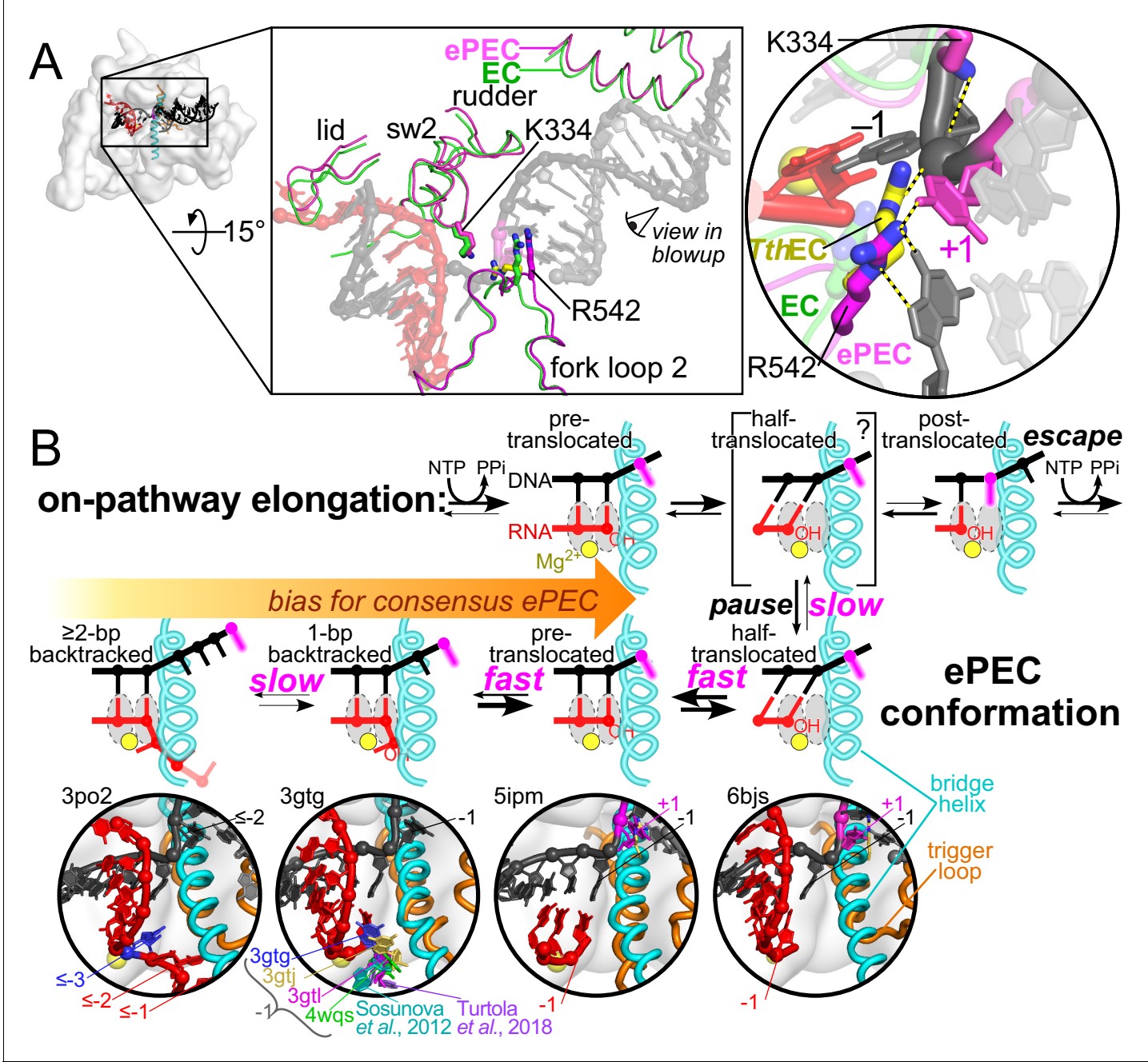

**Figure 8.** Multistate model of elemental pausing. (**A**) The small shift in RNAP modules observed in the ePEC (pdb 6bjs; magenta) relative to an EC (pdb 6alf; green) is depicted in the top central panel (*Kang et al., 2018a*; *Kang et al., 2017*). The locations of K334 in the EC (faint green) and ePEC (magenta) and the locations of R542 in a *Tth*RNAP EC (pdb 2o5i, yellow; *Vassylyev et al., 2007*), EC, and ePEC are shown in expanded view on the right. The +1 dC nucleotide trapped in the downstream DNA is magenta. (**B**) Discrete states present during formation of and escape from an elemental pause. The kinetic block is the loading of the template base, but the ePEC can assume other translocation registers depending on the sequences surrounding the pause. Examples of structures occupying the different translocation registers are shown in the close-ups below the active-site schematics (*Ó Maoiléidigh et al., 2011*; *Wang et al., 2009*; *Sosunova et al., 2013*; *Sekine et al., 2015*; *Liu et al., 2016*; *Kang et al., 2018a*; *Turtola et al., 2018*).

DOI: https://doi.org/10.7554/eLife.40981.020

makes the distinction between backtrack pausing and non-backtrack pausing a matter of degree rather than a clear-cut division.

Our results do not address the important question of whether the half-translocated state also is a significant kinetic intermediate during on-pathway nucleotide addition, although other studies suggest this possibility. Translocation can affect overall elongation rate (*Imashimizu et al., 2013*; *Gabizon et al., 2018*), and a half-translocated intermediate has been directly observed during *in crystallo* nucleotide addition by RNA-dependent RNAP (*Shu and Gong, 2016*). On-pathway translocation may proceed *via* initial RNA translocation then template DNA loading; the second step may be naturally slower at a pause signal, allowing time for formation of the ePEC conformation in which template-base loading becomes strongly inhibited (*Gabizon et al., 2018*).

The multistate model (*Figure 8*) has important regulatory implications. Just as different regulators can exert distinct effects on the multistep mechanism of transcription initiation by stabilizing or destabilizing different intermediates (*Hubin et al., 2017*), the existence of multiple elemental pause states affords multiple targets for regulators even when these intermediates are in equilibrium. For example, ppGpp is known to stimulate pausing (*Kingston et al., 1981*) and to promote backtracking (*Kamarthapu et al., 2016*); ppGpp could stimulate pausing by stabilizing a pretranslocated or backtracked PEC. Further, ≥1 bp backtracking may become more significant for other sequence contexts, conditions, or RNAPs (e.g., eukaryotic RNAPII).

## Multipartite RNAP contacts, not RNA-DNA energetics alone, drive elemental pausing

Our results (*Figure 2*) confirm that, like the hairpin-stabilized *his*PEC (*Chan et al., 1997*), the elemental pause signal is multipartite and involves significant contributions from the usFJ, hybrid, dsFJ, and dsDNA. Although strong evidence of significant contributions by the hybrid and dsDNA exists (*Bochkareva et al., 2012*; *Larson et al., 2014*; *Palangat and Landick, 2001*; *Palangat et al., 2004*), some descriptions of pausing focus only on the usFJ and dsFJ (*Vvedenskaya et al., 2014*; *Imashimizu et al., 2015*). The inhibitory contributions of the usFJ and dsFJ to hybrid unwinding and NTP binding, respectively, have been known since the earliest studies of pausing (*Gilbert et al., 1974*; *Aivazashvili et al., 1981*) but alone are inadequate to predict pause strength.

At least three factors may contribute to underestimation of the importance of hybrid and downstream DNA sequences in pausing. First, the widespread use of sequence logos to represent nucleic acid signals places undue weight on simple, independent interactions and underweights contributions of complex sequence interactions that affect energetics through nucleic-acid conformation or alternative side-chain contacts. Thus, a logo representing information content as single bp can cause complex, multipartite sequences to appear less important (*Figure 2A*). Although more sophisticated algorithms hold promise to characterize complex, multipartite signals like the elemental pause (*Siebert and Söding, 2016*), direct analyses of sequence variants remain the most reliable way to define relevant contributions. Direct mutational analyses establish the key contributions of the hybrid and downstream DNA in the elemental pause signal (*Figure 2*; *Bochkareva et al., 2012*; *Larson et al., 2014*; *Palangat and Landick, 2001*; *Palangat et al., 2004*).

Second, using NET-seq to identify a consensus pause signal requires rapid capture of PECs by quick-freezing actively growing cells in liquid $N_2$ (*Churchman and Weissman, 2012*; *Larson et al., 2014*). Quick-freezing reveals the modest sequence signatures of the hybrid and dsDNA in sequence logos and energetic analyses (*Figure 2A*; *Larson et al., 2014*). However, some studies that detected little if any hybrid or dsDNA contribution recovered cells by centrifugation before freezing, which may allow RNAP to escape all but the strongest pause sites and thus could over-represent contributions of the usFJ and dsFJ components (*Vvedenskaya et al., 2014*; *Imashimizu et al., 2015*).

Finally, the hybrid and dsDNA contact multiple RNAP modules (e.g., rudder, switch 2, clamp, *etc.*) and may principally affect the complex and modest conformational rearrangement into the ePEC state. Because this conformation is still incompletely understood, the contributions of the hybrid and dsDNA, which may involve specific duplex conformations, may be difficult to characterize.

Our findings taken together with earlier demonstrations of usFJ, hybrid, dsFJ, and dsDNA sequences that contribute to pausing should solidify a model of elemental pausing that depends on a multipartite pause signal.

## Structure of the elemental pause

The elemental pause is a distinct offline state, not an on-pathway elongation intermediate. Our study uncovered evidence for a modestly rearranged multistate ePEC in which the clamp is loosened and template-base loading is inhibited as well as a slower ePEC state entered by a minor but variable fraction of ECs (*Figures 1* and *2*, and *Figure 2—figure supplement 1*). The lack of effect of rapid GreA cleavage ruled out 1 bp backtracking as an explanation of the slow ePEC state, but the GreA/B experiments could not definitively rule out ≥2 bp backtracking because the maximal ≥3 nt cleavage rate (*Zhang et al., 2010*; *Sosunova et al., 2013*) is too close to slow ePEC escape rate. Alternatively or additionally, the slow ePEC state could involve RNAP swiveling, which is observed in the hairpin-stabilized PEC (*Kang et al., 2018b*). Swiveling involves a near-rigid body rotation of the clamp, shelf, SI3, and jaw, and inhibits nucleotide addition by a factor of ~10 by interfering with TL folding. Interestingly, the lifetime of the minor slow ePEC is ~10 times that of the majority ePEC state. Variations in susceptibility to swiveling, ≥2 bp backtracking, or both could explain why the slow fraction varies among RNAP preparations. Further studies will be needed to establish the structure and importance of the slow ePEC fraction, including detection with non-reconstituted transcription complexes or in vivo.

The multistate model of elemental pausing involving small changes in RNAP conformation and half-translocated, pretranslocated, and multiple 1 bp backtracked states described here will be difficult to test fully using ensemble biochemistry, single-molecule biochemistry, or crystallography. These methods are encumbered by the perturbing effects of probes and experimental configurations, the number of intermediates involved, and the rapid time-scale of their interchange. Time-resolved cryo-EM (*Frank, 2017*) promises an attractive approach if methods to distinguish intermediates during particle classification can be developed.

## Materials and methods

| Reagent type (species) or resource | Designation | Source or reference | Identifiers | Additional information |
|---|---|---|---|---|
| Strain, strain background (*E. coli*) | BL21λDE3 | Novagene-EMD Millipore | | expression strain |
| Recombinant DNA reagent | 6-MI-containing oligonucleotides | Fidelity Systems | | see *Supplementary file 1* for sequences |
| Recombinant DNA reagent | DNA and RNA oligos | Intergrated DNA Technologies (IDT) | | see *Supplementary file 1* for sequences |
| Chemical compound, drug | Cystamine dihydrochloride | MP Biomedicals | Cat# ICN10049205 | |
| Chemical compound, drug | Dithiothreitol (DTT) | Gold Biotechnology | Cat# DTT100 | |
| Chemical compound, drug | Iodoacetamide | Sigma-Aldrich | Cat# I6125 | |
| Chemical compound, drug | Heparin, Na salt | Sigma-Aldrich | Cat# H-3393 | |
| Chemical compound, drug | Ribonucleotide triphosphates | Promega | Cat# P1221 | grade I-A (181 USP units/mg) |
| Chemical compound, drug | Guanosine triphosphate (GTP) | GE Healthcare | Cat# 27-2076-01 | Low background flouresence |

*Continued on next page*

*Continued*

| Reagent type (species) or resource | Designation | Source or reference | Identifiers | Additional information |
|---|---|---|---|---|
| Chemical compound, drug | [γ-$^{32}$P]ATP | PerkinElmer Life Sciences | Cat# BLU002Z | |
| Chemical compound, drug | [α-$^{32}$P]CTP | PerkinElmer Life Sciences | Cat# BLU006H | |
| Chemical compound, drug | [α-$^{32}$P]GTP | PerkinElmer Life Sciences | Cat# BLU006H | |
| Chemical compound, drug | Acrylamide-Bisacrylamide | Bio-Rad | Cat# 1610145 | (19:1, 40% solution) |
| Software, algorithm | ImageJ | NIH (https://imagej.nih.gov/ij/) | | |
| Software, algorithm | Kaleidagraph | Synergy Software | | |
| Software, algorithm | Excel | Microsoft | | |
| Software, algorithm | ImageQuant | GE Healthcare | | |
| Software, algorithm | KinTek Explorer v6.1 | KinTek Corp. | | |
| Other | Phast gels | GE Healthcare | Cat# 17–067801 | |
| Other | HiTrap Heparin HP column | GE Healthcare | Cat# 17-0406-01 | |
| Other | HisTrap HP column | GE Healthcare | Cat# 17-5247-01 | |
| Other | Streptactin 5 ml High Capacity Column | IBA | Cat# 2-1238-001 | |

## Reagents and materials

Plasmids and oligonucleotides are listed in *Supplementary file 1*. RNA and DNA oligonucleotides were obtained from Integrated DNA Technologies (IDT; Coralville, IA) and purified by denaturing polyacrylamide gel electrophoresis (PAGE) before use. GreA, GreB, and RNAPs were purified as described previously (*Larson et al., 2014*; *Windgassen et al., 2014*). Briefly, His-tagged RNAPs were overexpressed in *E. coli* BL21 λDE3 and cells were lysed by sonication. RNAPs were enriched by PEI and ammonium sulfate precipitation, then purified by sequential nickel (5 mL HisTrap) and heparin (5 mL HiTrap) column chromatography, dialyzed into storage buffer (20 mM Tris-Cl, pH 8, 250 mM NaCl, 20 μM ZnCl$_2$, 1 mM MgCl$_2$, 0.1 mM EDTA, 1 mM DTT, and 25% glycerol), and stored in small aliquots at –80° C.

## In vitro transcription pause assays

PAGE-purified 15-mer RNA with 3′ end two nt upstream from the pause site (5 μM) and template DNA (10 μM) were annealed in transcription buffer 1 (TB1; 20 mM Tris-OAc pH 7.7, 5 mM Mg (OAc)$_2$, 40 mM KOAc, 1 mM DTT; sequences of RNAs and DNAs are in *Supplementary file 1*). Scaffolds were incubated with RNAP for 15 min at 37°C in TB1, then non-template DNA was added and incubation continued for 15 min at 37°C. The ratio of RNA:tDNA:RNAP:ntDNA was 1:2:3:5 (0.5 μM, 1 μM, 1.5 μM, 2.5 μM, respectively). ECs were diluted to 0.1 μM with TB1 + heparin (0.1 mg/ml), incubated for 3 min at 37°C, labeled by the incorporation of [α-$^{32}$P]GMP at 10 μM total GTP for 1 min at 37°C, and then placed on ice for 30–60 min. ECs were incubated for 3 min at 37°C before initiating the pause assay by addition of CTP to 100 μM and GTP to 10 or 100 μM in TB1 at 37°C. Reaction samples were removed at various time points and quenched with an equal volume of 2X stop

buffer (8 M urea, 50 mM EDTA, 90 mM Tris-borate buffer, pH 8.3, 0.02% each bromophenol blue and xylene cyanol). All remaining active ECs were chased to product by incubation with GTP at 1 mM for 1 min at 37°C. RNAs in each quenched reaction sample were separated by PAGE (15%; 19:1 acrylamide:bis-acrylamide) in 44 mM Tris-borate, pH 8.3, 1.25 mM $Na_2EDTA$, 8 M urea. The gel was exposed to a PhosphorImager screen, and the screen was scanned using Typhoon PhosphorImager software and quantified in ImageQuant. The averaged fraction of RNA at the position of the pause over time was fit to single- or double-exponential decay functions in KaleidaGraph to estimate pause efficiencies (amplitudes) and rate constants of pause escape. All pause kinetic parameters were determined from replicate (n $\geq$ 3) assays using error-weighted (SD) fits. For experiments comparing RNAPs, wild-type and variant RNAPs were purified side-by-side to avoid variable effects of different RNAP preparations on pausing kinetics.

## Rapid quench-flow measurement of nucleotide addition

To measure rates of C17 and G18 addition, G16 ECs were formed essentially as described for the pause transcription assay, but with 5'-[$^{32}$P]RNA limiting such that RNA:tDNA:RNAP:ntDNA was 1:1.3:2:3.3. To obtain nucleotide addition rates using a quench-flow apparatus (RQF-3; KinTek Corporation, Snow Shoe, PA), 400 nM G16 ECs were injected in one sample loop and 200 µM each CTP and GTP in TB1, in the other sample loop. Reactions were performed at 37°C for the designated times and quenched with 2 M HCl, then neutralized immediately to pH 7.8 with 3 M Tris base (supplemented with 250 µg torula yeast RNA/mL). RNA products were purified by phenol:chloroform extraction followed by ethanol precipitation, and resuspended in 1X stop buffer to a constant specific activity. RNA products from all timepoints were resolved by denaturing PAGE and quantified as described for pause transcription assays.

Reaction progress curves were generated for each RNA length (G16, C17$^+$, and G18$^+$) using KaleidaGraph (Synergy Software) by calculating the fraction of total RNA for each condition as a function of time. C17 and all RNAs longer than C17 were combined to give the C17$^+$ fraction; G18 and all RNAs longer than G18 were combined to give the G18$^+$ fraction. The averaged fraction at each time-point was then fit to a single-exponential equation.

## Kinetic modeling

To test whether the elemental pause is on online or offline state (i.e., involves a linear or branched kinetic mechanism; *Figure 1—figure supplement 1F,G*) and to test whether the slow fraction of ePECs was evident at only the first or at both pause sites on the tandem pause scaffold (*Figure 1—figure supplement 2*), we used kinetic modeling by numerical integration of pre-steady state rate equations using the program KinTek Explorer v6.1 (KinTek Corp., Snow Shoe, PA; *Johnson et al., 2009*). In both cases, to test whether the simpler mechanism was adequate to explain the data, we used replicate datasets (triplicate or greater) to generate a kinetic model for the rates of arrival at the pause site using the rate at which all RNAs before the pause site converted to RNAs at the pause site and beyond. We then held these rates constant and tested the simple kinetic models (linear, online pause (*Figure 1—figure supplement 1F,G*) or two populations of RNAP, fast and slow pausing (*Figure 1—figure supplement 2D,E*), including error for replicates to obtain the best fit and the residuals between the best fit and the observed data. We concluded that the more complex models (branched for *Figure 1—figure supplement 1F,G* or dynamic formation of the slow pause species for *Figure 1—figure supplement 2D,E*) were favored because the residuals for the simpler mechanism exhibited large, systematic variations whereas the more complex mechanisms exhibited smaller, random variations. We did not attempt to determine which mechanism best fit the data, and limited our conclusion to rejection of the simpler mechanism. For the dataset using the template from *Bochkareva et al. (2012)* (*Figure 1—figure supplement 1G*), for which we had six replicates, we determined error in the fits and residuals by individually fitting each dataset and calculating the average and error for the six fits.

## GreA- and GreB-stimulated cleavage assays

GreA- and GreB-stimulated cleavage and effects on pausing were assayed essentially as described earlier (*Larson et al., 2014*). Briefly, for pause assays (*Figure 1H* and *Figure 1—figure supplement 3B,C*), [α-$^{32}$P]GTP (10 µM) was maintained at constant specific activity throughout the labeling and

pause assay GreA, GreB, or both (1 μM each, final) were added concurrently with the CTP and UTP (100 μM each, final). The assays were performed in triplicate and analyzed as described above. To assay the rate of cleavage product accumulation (*Figure 1I* and *Figure 1—figure supplement 3B, D*), accumulating small RNAs were quantified from short phosphorimager screen exposures to avoid saturating the signal.

## Intrinsic cleavage assay

Intrinsic cleavage was assayed essentially as described earlier (*Mishanina et al., 2017*). Briefly, 3′-end labeled C17 ePECs (#9563 NT DNA, #8334 T DNA, #8401 RNA; *Supplementary file 1*) were formed and immobilized on Ni$^{2+}$-NTA beads, then washed to remove unincorporated [α-$^{32}$P]CTP. Cleavage was initiated at 37°C by resuspending washed ePECs with Cleavage Buffer (CB; 25 mM Tris·HCl pH 9.0, 50 mM KCl, 20 mM MgCl$_2$, 1 mM DTT, 5% glycerol, and 25 μg acetylated BSA/mL), and samples were collected at designated timepoints by mixing with 2X stop buffer. Cleavage products were separated by denaturing PAGE as described for transcription pause assays.

## Stopped-flow fluorescence translocation assay

To measure translocation rates of the hybrid (*Figure 3*), we used the assay developed by Belogurov and co-workers (*Malinen et al., 2012*; *Malinen et al., 2014*). PAGE-purified template DNA and RNA were annealed in TB1 (see *Supplementary file 1*). This scaffold was incubated with RNAP for 15 min at 37°C in TB, then non-template DNA was added such that the final ratio of tDNA:RNA:RNAP: ntDNA was 1:2:3:5 (2 μM: 4 μM: 6 μM: 10 μM, respectively). This solution was diluted to 0.4 μM RNAP with TB1.

ECs were then injected into one loading syringe of a stopped-flow apparatus (SF-300X; KinTek Corporation, Snow Shoe, PA) and 200 μM CTP in TB1 was loaded in the other syringe. Upon initiating rapid mixing at 37°C, 6-MI fluorescence was excited at 340 nm (2.4 nm bandwidth), and emission was monitored in real time through a 400 nm long-pass filter (Edmund Optics Inc., Barrington, NJ). The kinetics of 6-MI fluorescence unquenching was determined by fitting the average fluorescence (n ≥ 6 traces), normalized from 0 to 1, to a double exponential *Equation 1*:

$$F_t = A\left(1 - e^{-k_{1,obs}t}\right) + B\left(1 - e^{-k_{2,obs}t}\right) \qquad (1)$$

where, t = time (s); A = fast kinetic species signal amplitude; B = slow kinetic species signal amplitude; $k_{1,obs}$ = observed rate of fluorescence increase for the fast kinetic species; $k_{2,obs}$ = observed rate of fluorescence increase for the slow kinetic species. The fast species rate is reported as the translocation rate (*Malinen et al., 2012*).

## Equilibrium fluorescence measurement of translocation

To measure equilibrium translocation of downstream DNA (*Figure 4*), G16 ECs were formed in TB1 as described for the quench-flow experiment but without Mg(OAc)$_2$ to stabilize ECs and with the fluorescent ntDNA as the limiting component (ntDNA:tDNA:RNA:RNAP = 1:1.5:2:3 with ntDNA at 300 nM). For equilibrium measurements of hybrid translocation (*Figure 3*), G16 ECs were formed similarly but with the tDNA as the limiting component (see description in stopped-flow assay section). Fluorescence measurements were conducted using a PTI-spectrofluorometer (Model QM-4/2003, Photon Technology International) with 5 mM path length and 45 uL quartz cuvettes (Hellma Analytics, Müllheim, Germany). Emission spectra were obtained by exciting at 340 nm (5 nm bandwidth) and monitoring fluorescence between 360 and 500 nm (5 nm bandwidth).

For each substrate addition experiment, 60 μL EC was added to the cuvette and incubated for 2 min at 37°C in the cuvette holder before performing an emission scan (average of 3 traces at 0.25 nm step size). Then, NTPs were added (concurrently with 5 mM Mg(OAc)$_2$ plus additional Mg(OAc)$_2$ equal to the NTP concentration in the assay) and the incubation was continued for 1 min (5 min for 2′dCTP) at 37°C before performing an emission scan. Between fluorescence measurements, an aliquot was removed to 2X stop buffer to subjected to denaturing PAGE to confirm nucleotide addition. For the GTP titration condition, aliquots were only removed to confirm initial G16 RNA, 3′dC addition, and failure of 20 mM GTP to incorporate following 3′dC addition.

6-MI fluorescence was quantified at 425 nm. Background fluorescence from a GTP contaminant was subtracted using signal from GTP alone in buffer. We found that the level of fluorescence from

the contaminant varied significantly among vendors and GTP lot, with GTP from GE Healthcare containing the least amount. Reported increases in fluorescence are fold changes relative to the initial signal in G16 EC.

## Cys-pair reporter (CPR) crosslinking assays

CPR crosslinking assays (*Figure 5*) were performed as described previously (*Nayak et al., 2013*). Nucleic acid scaffolds were prepared by annealing RNA, template DNA (T-DNA) and 15 µM non-template DNA (NT-DNA) at 10 µM, 12 µM, and 15 µM final concentrations, respectively, in reconstitution buffer (RB; 20 mM Tris-HCl pH 8, 20 mM NaCl, and 1 mM EDTA). ECs, ePECs, and *his* PEC were formed by incubating 1 µM RNAP and scaffold (2 µM, based on RNA) in buffer A (50 mM Tris-HCl pH 8, 20 mM NaCl, 10 mM $MgCl_2$, 1 mM EDTA, and 2.5 ug acetylated bovine serum albumin/mL for 15 min at room temperature (RT). For crosslinking reactions with NTP, 3'deoxyECs formed by reaction with 3'dNTP were incubated for 15 min at RT with 0, 0.005, 0.01, 0.025, 0.05, 0.1, 0.25, 0.5, 1, 2.5, 5, and 10 mM GTP or ATP. Next, EC, ePEC, or his-PEC (final RNAP 0.8 uM and scaffold 1.6 uM) were incubated for 60 min with 2.5 mM CSSC and 0.05 mM DTT (E = –0.16) and stopped with 50 mM iodoacetamide. Samples were separated by native PAGE to verify reconstitution efficiency and by sodium dodecyl sulfate (SDS)-PAGE using 4–15% GE Healthcare PhastGel to quantify formation of crosslinks. Gels were stained with Coomassie Blue and imaged with a CCD camera. The fraction cross-linked was quantified with ImageJ software.

## Pause assays with CPR RNAPs

For crosslinked CPR transcription assays, nucleic-acid scaffolds containing RNA and template DNA (1:2 ratio of RNA to DNA) were used to reconstitute ePECs or control *his*PECs (*Figure 6C and D*) as described in *Kang et al., 2018a*. The U15 ePECs containing limiting CPR RNAP (1 µM) were reconstituted on 2 µM scaffold (based on RNA) for 15 min at 37°C in Elongation Buffer (EB; 25 mM HEPES-KOH, pH 8.0, 130 mM KCl, 5 mM $MgCl_2$, 1 mM DTT, 0.15 mM EDTA, 5% glycerol, and 25 µg acetylated bovine serum albumin/mL), followed by addition of 6 µM non-template DNA and further incubation for 10 min at 37°C to complete assembly of the transcription complexes. Wild-type RNAP was tested as a control side-by-side with CPR RNAPs. Crosslinking of 1 µM ePECs was performed in the presence of 1 mM cystamine as the oxidant and 0.8 mM DTT, for 15 min at 37°C. An aliquot of the crosslinking reaction was quenched with 15 mM iodoacetamide (final concentration) and analyzed by non-reducing SDS-PAGE to confirm formation of the crosslink.

The crosslinked U15 ePECs were diluted to 0.2 µM with EB (without DTT, for crosslinked samples) and incubated with heparin (0.1 mg/mL final) for 3 min at 37°C. The U15 ePECs were then radiolabeled by extension with 20 µM [α-$^{32}$P]GTP for 1 min at 37°C, to poise the complexes one nucleotide before the pause sequence. The resulting G16 ePECs were further diluted to 0.1 µM (based on RNAP) and assayed at 37°C for pause-escape kinetics at 10 µM GTP by addition of CTP in EB to 100 µM (without DTT, for crosslinked samples). Reaction samples were removed at various time points and quenched with an equal volume of 2X stop buffer. All active ePECs were chased out of the pause with 500 µM GTP and CTP, each, for 5 min at 37°C. RNAs in each quenched reaction sample were separated on a 15% urea-PAGE gel. Gels were visualized and quantified as described for in vitro transcription assays.

## Cys Triplet Reporter assays

For Cys triplet reporter (CTR) cross-linking assays (*Figure 6B,E and F*), ECs and PECs were assembled on purified DNA and RNA scaffolds specified in the figure legend and as described previously (*Kang et al., 2018a*). Briefly, 10 µM RNA, 12 µM template DNA, and 15 µM non-template DNA (*Supplementary file 1*) were annealed in RB. To assemble complexes, scaffold (2 µM) was mixed with limiting CTR RNAP (1 µM; CTR RNAP: β'1045iC 258iC, β843C) in 50 mM Tris-HCl, pH 7.9, 20 mM NaCl, 10 mM $MgCl_2$, 0.1 mM EDTA, 5% glycerol, and 2.5 µg of acetylated bovine serum albumin/mL, and added to mixtures of cystamine and DTT to generate redox potentials that ranged from −0.314 to −0.424. Complexes were incubated for 60 min at room temperature and then were quenched with the addition of iodoacetamide to 15 mM. The formation of cysteine-pair cross-links was then evaluated by non-reducing SDS-PAGE (4%–15% gradient Phastgel; GE Healthcare) as described previously (*Nayak et al., 2013*). Gels were stained with Coomassie Blue and imaged with

a CCD camera. The fraction cross-linked was quantified with ImageJ software. The experimental error was determined as the standard deviation of measurements from three or more independent replicates.

## Acknowledgements

We thank members of the Landick lab, Seth Darst, Jin Young Kang, and Georgiy Belogurov for many helpful discussions during the course of this work and preparation of the manuscript. The work was supported by a grant from the NIH to RL (R01 GM38660). TVM was supported by a Career Award at the Scientific Interface from the Burroughs Wellcome Fund.

## Additional information

### Funding

| Funder | Grant reference number | Author |
| --- | --- | --- |
| National Institute of General Medical Sciences | R01 GM38660 | Robert Landick |

The funders had no role in study design, data collection and interpretation, or the decision to submit the work for publication.

### Author contributions

Jason Saba, Conceptualization, Investigation; Xien Yu Chua, Tatiana V Mishanina, Dhananjaya Nayak, Tricia A Windgassen, Investigation; Rachel Anne Mooney, Supervision, Investigation, Writing—review and editing; Robert Landick, Conceptualization, Formal analysis, Supervision, Funding acquisition, Investigation, Writing—original draft, Project administration, Writing—review and editing

### Author ORCIDs

Jason Saba 🔟 https://orcid.org/0000-0001-7273-2970
Xien Yu Chua 🔟 https://orcid.org/0000-0003-2455-9493
Robert Landick 🔟 http://orcid.org/0000-0002-5042-0383

### Decision letter and Author response

Decision letter https://doi.org/10.7554/eLife.40981.024
Author response https://doi.org/10.7554/eLife.40981.025

## Additional files

### Supplementary files

• Supplementary file 1. Oligonucleotides and plasmids used in this study.
DOI: https://doi.org/10.7554/eLife.40981.022
• Transparent reporting form
DOI: https://doi.org/10.7554/eLife.40981.023

### Data availability

All data generated or analysed during this study are included in the manuscript and supporting files. Source data files have been provided for Figures 2 and 7.

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
