## [Decision Letter]

Thank you for submitting your article "The elemental mechanism of transcriptional pausing" for consideration by *eLife*. Your article has been reviewed by two peer reviewers, and the evaluation has been overseen by Jerry Workman as Reviewing Editor and James Manley as the Senior Editor. One of the two reviewers has agreed to reveal their identity: Dong Wang.

The reviewers have discussed the reviews with one another and the Reviewing Editor has drafted this decision to help you prepare a revised submission.

Summary:

This comprehensive study is focused on understanding the fundamental mechanism of the elemental paused state during bacterial transcription. Several longstanding unresolved issues related to elemental pausing include: whether the elemental paused elongation complex (ePEC) is an on-pathway state or an offline state; whether the elemental pause is non-backtracked or backtracked; what is the nature the elemental pause signal? Using multiple biochemical (kinetic) and biophysical approaches, the authors revealed that the RNAP elemental pause state is an off-pathway with a branched kinetic mechanism. While previous CryoEM structures showed that the ePEC is in a half-translocated state such that RNA is translocated but the template is not loading, the current authors found that this ePEC can easily enter pretranslocated and one base-pair backtracked states. Furthermore, the authors revealed that the RNAP elemental pause signal is multipartite. Taken together, the authors proposed a multistate model that provides an important framework to understand how transcriptional pausing is modulated. The data are solid and the manuscript is well-written. It will be of general interest for the transcription field and readers of e*Life*.

We only have a few minor comments:

"the bypass fraction was shifted from ~22% at 10 μM GTP to ~7% at 100 μM GTP". Why is the bypass fraction lower at higher GTP here? This is inconsistent with Figure 1—figure supplement 1C that bypass fraction increases at higher GTP.

"upstream fork-junction (usFJ" should be "upstream fork-junction (usFJ)".

Subsection “Hybrid translocation is not rate-limiting for elemental pause escape”, first paragraph; please change to "(Figures 3D-F)", as Figure 3F is not cited in the manuscript.

Subsection “Hybrid translocation is not rate-limiting for elemental pause escape”, first paragraph: the reference should be (Malinen et al., 2012) instead.

"control EC (Figure 4C; (Malinen et al., 2014)." should be "control EC (Figure 4C; Malinen et al., 2014)."

"(compare to control EC; Figure 4C, D, E)" it shall be Figure 5C, D, E instead?

Subsection “Clamp loosening but not extensive clamp opening accompanies elemental pausing”, first paragraph: "Figure 5D". should be Figure 6D instead.

"apparent H-bonds of R542 in fork loop 2 to the template dC that may hinder its translocation into the active site". Could the authors comment whether R452 can interact with other +1 template bases, is this interaction sequence-specific?

"15% PAG" should be "15% PAGE".

---

## [Author Response]

We only have a few minor comments:"the bypass fraction was shifted from ~22% at 10 μM GTP to ~7% at 100 μM GTP". Why is the bypass fraction lower at higher GTP here? This is inconsistent with Figure 1—figure supplement 1C that bypass fraction increases at higher GTP.

We thank the reviewers for catching this error. Bypass indeed increases at higher GTP, as shown in Figure 1G also. The sentence in question contained a typo reversing the GTP concentrations. It is now corrected in the revised manuscript to read "the bypass fraction was shifted from ~22% at 100 µM GTP to ~7% at 10 µM GTP".

"upstream fork-junction (usFJ" should be "upstream fork-junction (usFJ)".

We thank the reviewers for catching this error and have made the suggested correction.

Subsection “Hybrid translocation is not rate-limiting for elemental pause escape”, first paragraph; please change to "(Figures 3D-F)", as Figure 3F is not cited in the manuscript.

We thank the reviewers for catching this error and have made the suggested correction.

Subsection “Hybrid translocation is not rate-limiting for elemental pause escape”, first paragraph: the reference should be (Malinen et al., 2012) instead.

We thank the reviewers for catching this error and have made the suggested correction.

"control EC (Figure 4C; (Malinen et al., 2014)." should be "control EC (Figure 4C; Malinen et al., 2014)."

We thank the reviewers for catching this error and have made the suggested correction.

"(compare to control EC; Figure 4C, D, E)" it shall be Figure 5C, D, E instead?

We thank the reviewers for catching this error and have made the suggested correction.

Subsection “Clamp loosening but not extensive clamp opening accompanies elemental pausing”, first paragraph: "Figure 5D". should be Figure 6D instead.

We thank the reviewers for catching this error and have made the suggested correction.

"apparent H-bonds of R542 in fork loop 2 to the template dC that may hinder its translocation into the active site". Could the authors comment whether R452 can interact with other +1 template bases, is this interaction sequence-specific?

We appreciate this suggestion from the reviewers. The low resolution of the ePEC cryo-EM structure (5.5 Å) makes discussion of details of the Arg-dC interaction speculative (we note this in the Results section). The interaction is potentially specific for the dC–dG as contact to the nontemplate G (H-bonds to O6 and N7) are possible, and this pattern is G-specific. The Arg contact to the WC face of template dC seen in the Murakami crystal structure or an initiation (5voi) also is cytosine-specific. Thus, it is possible that R542 may inhibit translocation by more than one type contact specific to a +1 dC in the template strand. Because of structural uncertainties, we have limited the description of the possible contacts to addition of a single sentence: “Notably the apparent R542 H-bonding contacts to the +1 dC–dG bp as well as to an unpaired but pretranslocated +1 dC in an initiation complex (Murakami et al., 2017) are not feasible for other +1 bases, possibly suggesting multiple ways R542 could inhibit template DNA translocation in an ePEC.”. We thank the reviewers for suggesting this addition, which should help readers better put our results into context. We also have edited the description of the R542 interaction in the results to refer to the dC–dG bp instead of simply to template dC and we have modified Figure 7B to include depiction of the potential H-bond to N7 (2.9 Å in 6bjs).

"15% PAG" should be "15% PAGE".

We appreciate that PAGE (polyacrylamide gel electrophoresis) is the more commonly recognized abbreviation, although in this instance PAG was an abbreviation for polyacrylamide gel not including electrophoresis. Since all other usages of this abbreviation in the manuscript are PAGE, we revised the sentence to now read “RNAs in each quenched reaction sample were separated by PAGE (15%; 19:1 acrylamide:bis-acrylamide)…”